# Peroxisomes contribute to intracellular calcium dynamics in cardiomyocytes and non-excitable cells

Yelena Sargsyan[1], Uta Bickmeyer[1], Christine S Gibhardt[2], Katrin Streckfuss-Bömeke[3,4,6], Ivan Bogeski[2], Sven Thoms[1,5,6]

Peroxisomes communicate with other cellular compartments by transfer of various metabolites. However, whether peroxisomes are sites for calcium handling and exchange has remained contentious. Here we generated sensors for assessment of peroxisomal calcium and applied them for single cell-based calcium imaging in HeLa cells and cardiomyocytes. We found that peroxisomes in HeLa cells take up calcium upon depletion of intracellular calcium stores and upon calcium influx across the plasma membrane. Furthermore, we show that peroxisomes of neonatal rat cardiomyocytes and human induced pluripotent stem cell–derived cardiomyocytes can take up calcium. Our results indicate that peroxisomal and cytosolic calcium signals are tightly interconnected both in HeLa cells and in cardiomyocytes. Cardiac peroxisomes take up calcium on beat-to-beat basis. Hence, peroxisomes may play an important role in shaping cellular calcium dynamics of cardiomyocytes.

## Introduction

Calcium ions ($Ca^{2+}$) play a decisive role in the regulation of many cellular processes and inter-compartment communication, especially in excitable cells such as neurons or cardiomyocytes (CMs) (Clapham, 2007). In CMs, for example, cytosolic $Ca^{2+}$ directly engages in cell contraction. At the same time, mitochondrial $Ca^{2+}$ coordinates ATP production and energy demand in CMs (Williams et al, 2015), highlighting the importance of intracellular organelles in $Ca^{2+}$ redistribution. The main sites of $Ca^{2+}$ entry to the cell and intracellular calcium signal regulation are the plasma membrane (PM) and intracellular calcium stores, in particular those of the ER (Paupe & Prudent, 2018).

Excess of organellar $Ca^{2+}$ can be detrimental for health. Elevated mitochondrial uptake increases mitochondrial reactive oxygen species (ROS) production and is associated with heart failure and ischemic brain injury (Starkov et al, 2004; Santulli et al, 2015). Reversely, mitochondrial ROS decrease if $Ca^{2+}$ uptake to mitochondria

is suppressed (Mallilankaraman et al, 2012; Tomar et al, 2016). Understanding principles and mechanisms of organellar $Ca^{2+}$ handling provides a starting point to develop interventions in dysregulated calcium handling.

Peroxisomes are small intracellular organelles with a phospholipid bilayer membrane. In concert with evolutionarily conserved functions in lipid and redox metabolism, peroxisomes are highly plastic and change in their number, morphology and content upon environmental stimuli (Smith & Aitchison, 2013). Communication of peroxisomes with other cellular compartments through exchange of ROS or lipid metabolites is essential for human health (Wanders et al, 2015; Castro et al, 2018; Schrader et al, 2020). Yet, peroxisomal $Ca^{2+}$ has not been studied in excitable cells before, and there are contradicting data about the $Ca^{2+}$ handling in peroxisomes and its dependence on cytosolic $Ca^{2+}$ (Drago et al, 2008; Lasorsa et al, 2008). It has been suggested that peroxisomes are potential targets of $Ca^{2+}$ signalling pathways that initiate outside of the peroxisome or serve as a cytosolic $Ca^{2+}$ buffer, but peroxisomes may also take up $Ca^{2+}$ due to their own need (Drago et al, 2008; Islinger et al, 2012).

Measurement of $Ca^{2+}$ dynamics in vivo inside cellular organelles was driven by the development of $Ca^{2+}$-sensitive fluorescent proteins, also known as genetically encoded $Ca^{2+}$ indicators (GECIs) (Pozzan & Rudolf, 2009; Gibhardt et al, 2016). $Ca^{2+}$ dynamics was analysed in the ER, in mitochondria, the cytosol, and in lysosomes by using GECIs (Whitaker, 2010; McCue et al, 2013). GECIs have a $Ca^{2+}$-binding domain, usually CaM. Ratiometric pericam is a single fluorophore-based GECI with circularly permuted EYFP (cpEYFP) as fluorophore (Nagai et al, 2001). GECIs play a special role among cameleon-based sensors that use Förster resonance energy transfer (FRET). Here, $Ca^{2+}$ results in a conformational change that decreases the distance between donor (typically CFP) and acceptor (typically a YFP variant) enabling FRET (Palmer & Tsien, 2006; Pérez Koldenkova & Nagai, 2013; Gibhardt et al, 2016) (Fig 1A).

Patients with adult Refsum disease due to peroxisome biogenesis disorder develop cardiac arrhythmias and heart failure at advanced disease stages (Wanders & Komen, 2007). As the heart muscle uses fatty acids as its main energy source, peroxisome

[1]Department of Child and Adolescent Health, University Medical Center, Göttingen, Germany   [2]Molecular Physiology, Institute of Cardiovascular Physiology, University Medical Center, Göttingen, Germany   [3]Clinic for Cardiology and Pneumology, University Medical Center, Göttingen, Germany   [4]Institute of Pharmacology and Toxicology, Würzburg University, Würzburg, Germany   [5]Department of Biochemistry and Molecular Medicine, Medical School, Bielefeld University, Bielefeld, Germany   [6]German Center of Cardiovascular Research (DZHK), Partner Site Göttingen, Germany

Correspondence: sven.thoms@med.uni-goettingen.de; sven.thoms@uni-bielefeld.de

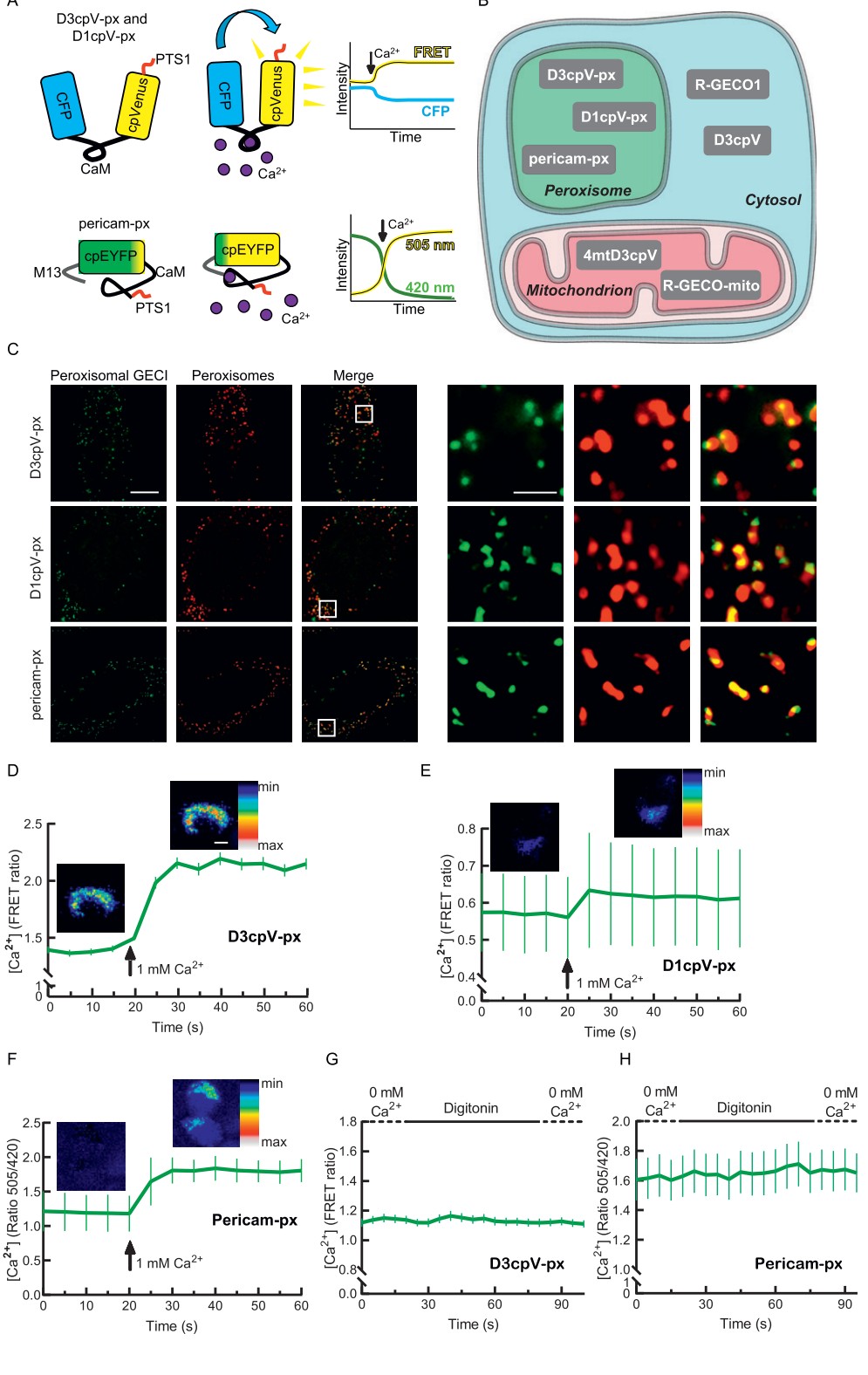

**Figure 1. New sensors for peroxisomal Ca²⁺.**

**(A)** Genetically encoded calcium indicators (GECIs) targeted to peroxisomes. D3cpv-px and D1cpv-px are Förster resonance energy transfer (FRET) sensors with modified CaM sites. Pericam-px is a single fluorophore-based GECI that has M13 and CaM as $Ca^{2+}$-binding sites. In the absence of $Ca^{2+}$, the emission measured when the sensor is excited with 420 nm is higher than when excited with 505 nm. The ratio 505/420 is a measure for the $Ca^{2+}$ concentration. **(B)** Subcellular localisation of GECIs used in this study. **(C)** Peroxisomal GECIs colocalise with the peroxisomal membrane marker PEX14 or PMP70. HeLa cells were transfected with the GECIs and stained with anti-PEX14 or anti-PMP70 antibodies. The images in the left part of the panel show one cell each (scale bar 10 μm). The cropped areas are marked and magnified in the right part of the panel (scale bar 2 μm). **(D, E, F)** D3cpv-px, D1cpv-px, and pericam-px are $Ca^{2+}$ sensitive. Images false-colored with look-up table show representative cells before (left) and after (right) $Ca^{2+}$ addition. Curves presented as mean ± SEM. Scale bar: 10 μm. **(D)** Addition of 1 mM $Ca^{2+}$ to D3cpV-px expressing cells results in 1.5-fold FRET ratio increase, n = 60 cells from three independent experiments. **(E)** FRET ratio increases 1.08 times when 1 mM $Ca^{2+}$ is added to D1cpv-px expressing cells, n = 33 cells from three experiments. **(F)** $Ca^{2+}$ addition leads to 1.5-fold increase in 505/420 ratio with pericam-px, n = 75 cells from three experiments. **(G)** Measurement of D3cpV-px during cytosol washout. No change in signal is detected. **(H)** Measurement of pericam-px during cytosol washout. **(G)** No difference of signal before and after cytosol washout is detected, n = 43 cells for D3cpV-px in (G) and n = 45 cells for pericam-px in (H).

localised lipid metabolism is thought to be especially important for the heart (Colasante et al, 2015). The cardioprotective effects of peroxisomes are also attributed to their role in redox homeostasis (Colasante et al, 2015). However, if peroxisomes directly participate in cellular $Ca^{2+}$ homeostasis, they may have also antiarrhythmic effects independent of their metabolic roles.

This work combines the advantages of organelle-targeted GECIs and human induced pluripotent stem cells (hiPSCs). We develop

several peroxisomal Ca$^{2+}$ sensors, and we measure intraperoxisomal Ca$^{2+}$ after pharmacological stimulation in non-excitable and excitable cells. We show that peroxisomes take up Ca$^{2+}$ upon cytosolic Ca$^{2+}$ increase after both ER Ca$^{2+}$-store depletion and Ca$^{2+}$ entry to the cells through PM. We also demonstrate that peroxisomes take up Ca$^{2+-}$ in rat CMs and hiPSC-CMs.

## Results

### Development and validation of Ca$^{2+}$ sensors for peroxisomal Ca$^{2+}$

To assess peroxisomal Ca$^{2+}$, we used three GECIs with different affinities to Ca$^{2+}$: D3cpV, D1cpV, and ratiometric pericam (Fig 1A and Table 1). The sensors were chosen to cover a wide range of K$_d$ values to identify the most suitable GECI for intraperoxisomal measurement. We preferred ratiometric sensors that allow measurements in two wavelengths. This enables direct interpretation of the acquired data by calculating the ratio of intensities at each time point. The ratios provide direct information about Ca$^{2+}$ concentration and are independent of the sensor expression itself (Pérez Koldenkova & Nagai, 2013). For the straight comparison of cellular compartments, we used specific sensors for the cytosol (D3cpV and R-GECO1), mitochondria (4mtD3cpV and R-GECO-mito), and peroxisomes (D3cpV-px, D1cpV-px, and pericam-px) (Fig 1B).

D3cpV is a cameleon-type indicator based on FRET. The conformational change associated with the Ca$^{2+}$ binding to CaM leads to an increase in FRET efficiency and FRET ratio (Pérez Koldenkova & Nagai, 2013). D3cpV has an in vitro K$_d$ value of 0.6 µM and a dynamic range of 5.0 (Palmer et al, 2006). D1cpV, in comparison, is a FRET sensor with a K$_d$ value of 60 µM (Palmer et al, 2004). Finally, pericam is a cpEYFP-based GECI with two excitation peaks at ~420 and ~505 nm (Nagai et al, 2001). In the presence of Ca$^{2+}$, a conformational change in the pericam structure shifts the excitation profile so that the 505/420 ratio increases and serves as a measure of Ca$^{2+}$ concentration (Fig 1A). Pericam has a K$_d$ value of 1.7 µM and dynamic range of 10.

We added strong peroxisomal targeting signals of the PTS1 type to D3cpV, D1cpV, and pericam and tested their localisation after transfection by co-staining with antibodies directed against the peroxisomal membrane protein PEX14. All constructs targeted to peroxisomes (Fig 1C).

To test if D3cpV-px senses Ca$^{2+}$ in peroxisomes of living cells, we permeabilized cells by digitonin, washed out the cytosol, and added

1 mM Ca$^{2+}$. Ca$^{2+}$ addition resulted in drastic increase of FRET and a 1.5-fold increase in FRET ratio (Fig 1D). To illustrate the increase of the FRET signal, we false-colored the images recorded before and after Ca$^{2+}$ addition by using a color look-up table (LUT) (Fig 1D).

When we performed the same type of experiment with D1cpV-px, FRET increased as well after Ca$^{2+}$ addition, showing that the D1cpV-px construct is Ca$^{2+}$ sensitive (Fig 1E). However, following the same stimulation protocol, the signal change of D1cpV-px was only 1.08-fold, and thus considerably smaller than for D3cpV-px. Because of the low signal change, we excluded D1cpV-px from further experiments on peroxisomal Ca$^{2+}$. Using pericam-px, the third peroxisome-targeted sensor in this set of experiments, high concentration of Ca$^{2+}$ addition after digitonin treatment resulted in 1.5-fold increase similar as for D3cpv-px (Fig 1F). Based on these results we decided to use D3cpv-px and pericam-px to evaluate Ca$^{2+}$ dynamics in peroxisomes.

To study possible mislocalisation or residual signal of peroxisomal Ca$^{2+}$ sensors from the cytosol, we analysed the peroxisomal Ca$^{2+}$ signals after digitonin permeabilisation of intact cells. If the sensor was partially mislocalised to the cytosol, we would expect a signal decrease after permeabilisation with digitonin. We first tested this in D3cpV-px (Fig 1G). There was no signal change observed, suggesting that D3cpV-px has no cytosolic mislocalisation. The cytosol washout also did not change the Ca$^{2+}$ signal of the pericam-px before and after digitonin treatment, suggesting that pericam-px, like D3cpV-px, is exclusively localised to the peroxisome (Fig 1H). The quantification of D3cpV-px colocalisation with peroxisomal enzyme catalase in comparison with peroxisomal membrane protein PMP70 and catalase colocalisation revealed no differences, suggesting residue-free targeting of the GECI to peroxisomes (Fig 2A).

### Peroxisomal Ca$^{2+}$ in non-excitable cells largely follows cytosolic Ca$^{2+}$

We first aimed to exclude that potential differences of cytosolic and peroxisomal pH lead to differences in the performance of our GECIs in these cell compartments. Therefore, we exposed the cells transfected with either D3cpV or with D3cpV-px to different pH buffers containing the proton ionophore nigericin (Fig 2B). In the range of physiological cytosolic pH (7.2, Casey et al, 2010) and peroxisomal pH (between 7.0 and 8.0, depending on the cell type and source, Godinho & Schrader, 2017), the sensors showed stable FRET ratios. As a control, we also exposed the sensors to pH 4.0,

**Table 1.   Key properties of the genetically encoded Ca$^{2+}$ indicators (GECIs) for cytosol and peroxisome.**

| Cytosolic GECIs | | | Peroxisomal GECIs (this study) | |
|---|---|---|---|---|
| Construct Name | K$_d$ (in vitro) | Dynamic range, D | Construct Name | Maximal increase upon 1 mM Ca$^{2+}$ addition |
| D3cpV | 0.6 µM[a] | 5.0[a] | D3cpV-px | 1.50× |
| D1cpV | 60 µM[b] | 1.7[c] | D1cpV-px | 1.08× |
| Ratiometric-pericam | 1.7 µM[d] | 10.0[d] | Pericam-px | 1.50× |

[a]References: Palmer et al (2006).
[b]Palmer et al (2004).
[c]Greotti et al (2016).
[d]Nagai et al (2001).

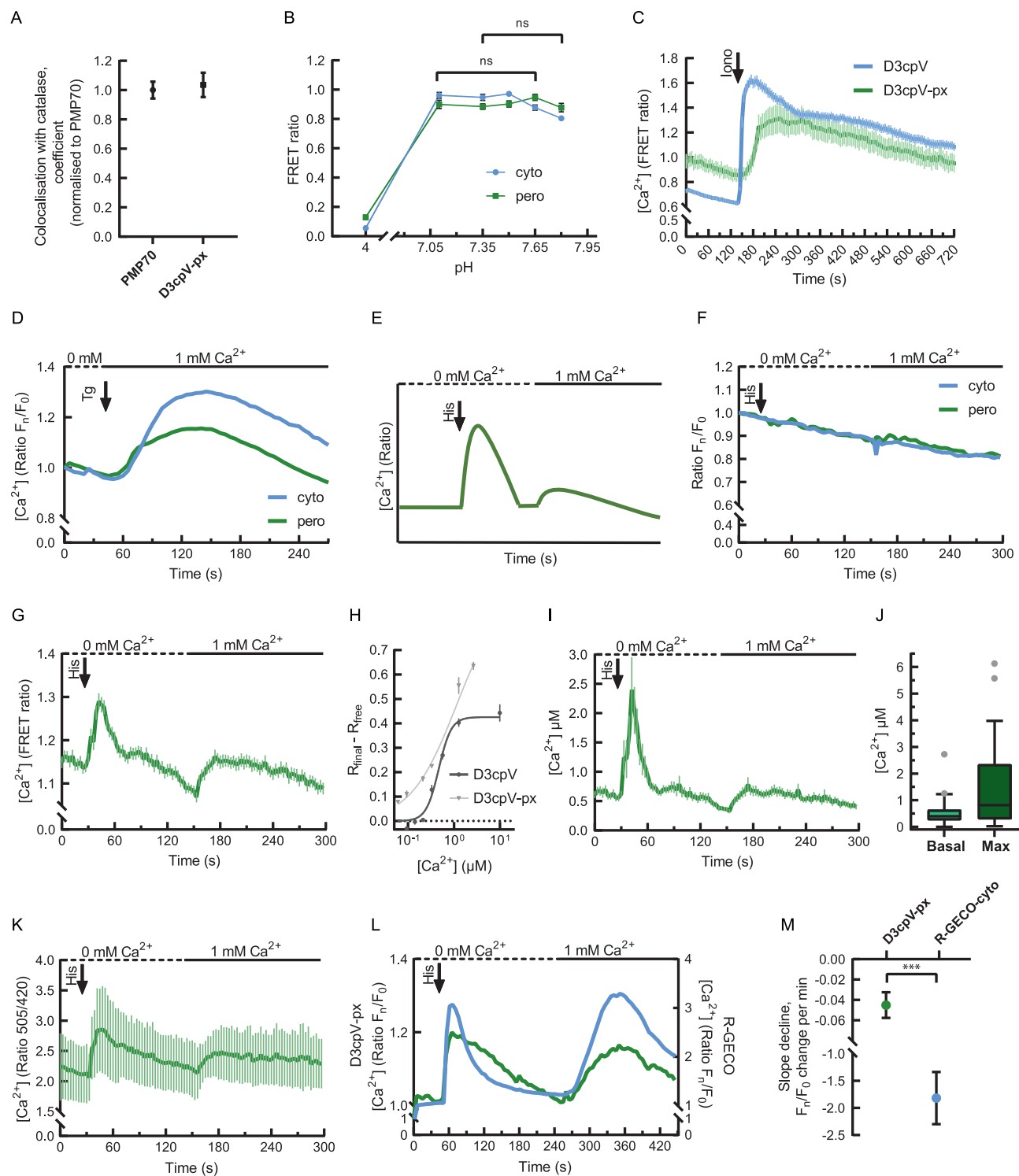

**Figure 2.  Measurement of peroxisomal Ca²⁺ in HeLa cells.**

**(A)** Quantification of D3cpV-px colocalisation with peroxisomes. Mander's colocalisation coefficient was normalised to PMP70 and catalase colocalisation, which was set to 1. n = 5. **(B)** Förster resonance energy transfer ratio measured at different pH values for D3cpV (cyto) and D3cpV-px (pero). Cells were incubated in buffers with different pH values containing 10 μM nigericin. Cyto and pero show comparable results at physiological pH values of cytosol and peroxisomes. At the pH = 4 the Förster resonance energy transfer ratio decreases drastically because of the acceptor sensitivity. Cell numbers for cyto at pH 4 n = 51, 7.1 = 51, 7.35 = 51, 7.5 = 67, 7.65 = 48, 7.8 = 48; for pero at pH 4 n = 50, 7.1 = 50, 7.35 = 71, 7.5 = 71, 7.65 = 64, 7.8 = 64 from three independent experiments per condition. **(C)** Comparison of cytosolic and peroxisomal responses to ionomycin (Iono). In comparison to cytosol, peroxisomal signal increases gradually, n = 16 cells for D3cpV and n = 9 cells for D3cpV-px. **(D)** One-step experiment in HeLa cells with thapsigargin (Tg) addition in Ca²⁺-containing buffer. Cytosolic and peroxisomal Ca²⁺ increase upon Tg treatment. n = 31 (cyto), 30 (pero) from four (cyto) and five (pero) independent experiments. **(E)** Experimental paradigm of a two-step Ca²⁺ measurement in non-excitable cells. First peak after histamine

which resulted in drastic decrease in FRET ratio due to the pH sensitivity of the acceptor (Fig 2B).

We then aimed to compare the maximal possible response of cytosol and peroxisomes to $Ca^{2+}$. For this purpose, we used ionomycin as an ionophore. Ionomycin resulted in fast and immediate increase in the cytosolic signal (Fig 2C). The peroxisomal signal also increased, yet more slowly. After reaching its maximum, it decreased gradually and in 12 min nearly returned to its starting values. The cytosolic signal decreased to its half maximal response in the same time with the most significant decrease observed in the first 2 min after the maximum (Fig 2C). These observations suggest that there could be differences between peroxisome and cytosol in $Ca^{2+}$ handling also under near-physiological stimulation.

Furthermore, we compared the response of cytosol and peroxisomes to milder cytosolic $Ca^{2+}$ increase stimulated with thapsigargin (Tg) as a sarcoplasmic/endoplasmic reticulum calcium ATPase (SERCA) inhibitor (Fig 2D). SERCA pumps $Ca^{2+}$ constantly back to the ER store (Clapham, 2007), upon its inhibition $Ca^{2+}$ accumulates in the cytosol. When cytosolic $Ca^{2+}$ increased gradually, we observed a difference only in maximal signals between peroxisome and cytosol.

Based on the $Ca^{2+}$ measurements in other organelles (Zhao et al, 2011; Matsuda et al, 2013; Suzuki et al, 2014; Petrungaro et al, 2015), we developed an experimental paradigm for peroxisome responses to the depletion and refilling of intracellular $Ca^{2+}$ stores, specifically ER, in non-excitable HeLa cells at near-physiological situation (Fig 2E). The stimulation of cell–surface localised G-protein–coupled receptors by 100 μM histamine results in the activation of phospholipase C cascade. Inositol 1,4,5-trisphosphate ($IP_3$), the product of the cascade, binds to the $IP_3$ receptor on the ER membrane, triggering $Ca^{2+}$ store release. The cells are then exposed to 1 mM extracellular $Ca^{2+}$, which leads to store-operated $Ca^{2+}$ entry and a second $Ca^{2+}$ elevation in the cytosol.

To confirm that this stimulation protocol does not result in changes of the acceptor fluorophore (YFP or Venus) signal of the GECI (e.g., due to drastic changes in pH during experiments), we applied the treatment protocol from Fig 2E on cells transfected with the cytosolic or the peroxisomal GECI acceptor in the absence of donor or $Ca^{2+}$-binding site. Neither histamine addition nor addition of 1 mM $Ca^{2+}$ showed signal change. We detected, however, mild photobleaching over time (Fig 2F).

When we treated HeLa cells expressing D3cpV-px according to this protocol, we observed two peaks (Fig 2G). Histamine addition resulted in a steep and fast increase in intraperoxisomal $Ca^{2+}$ based on depletion of the ER. Addition of extracellular $Ca^{2+}$ resulted in more gradual increase and gradual return to basal levels (Fig 2G).

For the interpretation of absolute peroxisomal $Ca^{2+}$ concentrations and their comparison with the cytosolic concentrations we performed in situ titration experiments. The results were fit into the one-site binding model with Hill coefficient (Palmer et al, 2006; Park & Palmer, 2015) (Fig 2H). We found the $K_d$ of D3cpV to be 0.47 μM, which is very close to the in vitro value 0.6 μM. The $K_d$ of D3cpV-px was 1.1 μM and only slightly higher than the cytosolic. Using the measurements with D3cpV-px, the known properties of the sensor and the measured $K_d$ values for D3cpV-px in our experimental settings, we calculated the absolute $Ca^{2+}$ concentration (Fig 2I) applying the formula described by Palmer and Tsien (2006). We find that under basal conditions, $Ca^{2+}$ level in peroxisomes is around 600 nM and it rises upon near-physiological stimulation with histamine up to 2.4 μM $Ca^{2+}$ (Fig 2J). The $Ca^{2+}$ dynamics in peroxisomes measured with D3cpV-px was reproduced by pericam-px: a larger peak is observed after ER-store depletion and a smaller one after extracellular $Ca^{2+}$ addition. The observed ratio curve from pericam-px largely resembles that from D3cpV-px. Because pericam has a reported in vitro $K_d$ value of 1.7 μM and covers higher $Ca^{2+}$ concentrations, the observed result confirms the upper limit of peroxisomal $Ca^{2+}$ and the range of $Ca^{2+}$ between 0.6 and 2.4 μM (Fig 2K). For better consistency and comparability of the results, and because pericam is described as more pH sensitive than D3cpV (Nagai et al, 2001), we decided to perform all further experiments with D3cpV-px.

To confirm that the response in our experiments is due to the immediate increase in $Ca^{2+}$ concentration, and to be able to directly compare peroxisomal $Ca^{2+}$ handling with that of the cytosol, cells were co-transfected with D3cpV-px and the mApple-based cytosolic $Ca^{2+}$ sensor R-GECO1, which increases in intensity when binding $Ca^{2+}$ (Zhao et al, 2011). A large increase in the red signal from R-GECO1 was observed upon both ER-store depletion and addition of extracellular $Ca^{2+}$ (blue curve in Fig 2L). Although the GECIs used for the measurement in two compartments have different properties that can result in differences in their kinetics, peroxisomes largely follow the $Ca^{2+}$ changes in the cytosol. Interestingly, there is little or no delay between signal increase in cytosol and peroxisomes when stimulated with histamine, and the post-stimulation decline is more gradual and prolonged in peroxisomes compared with the cytosol, indicating the existence of a possible barrier or gate that can be saturated (Fig 2M).

To compare peroxisomal $Ca^{2+}$ levels at rest and under stimulation with that of cytosol and mitochondria, cells were transfected with D3cpV sensors targeting specifically these compartments. FRET ratio was assessed as a direct indicator of $Ca^{2+}$ concentration (Fig 3). All three compartments showed two peaks: one after ER-store depletion with histamine, and another after extracellular $Ca^{2+}$ addition and PM-based uptake (Fig 3A).

The basal levels of $Ca^{2+}$ in mitochondria and peroxisomes detected with this sensor were comparable and significantly higher than that in the cytosol (typically ≈100 nM, Paupe & Prudent, 2018) in

(His) addition: ER-store depletion. Second peak, after addition of extracellular $Ca^{2+}$: plasma membrane-based uptake. **(F)** Measurement of the effect of treatments from (E) on the acceptor. Neither cytosolic EYFP (cyto) nor peroxisomal Venus-ACOX3 (pero) undergo changes upon treatment, suggesting that the pH changes during the experiment will not affect the measurement with genetically encoded $Ca^{2+}$ indicators. Data presented as mean, n = 44 (cyto), 44 (pero). **(G)** Measurement with D3cpV-px according to the paradigm in (E). Two $Ca^{2+}$ peaks of the experimental paradigm are detectable with D3cpV-px, n ≥ 50 cells from three experiments. **(H)** In situ $K_d$ calculation and the saturation curve of D3cpV and D3cpV-px. Curve fitting was performed with one-site model with Hill coefficient. **(G, I)** Absolute $Ca^{2+}$ concentration dynamics calculated from the data in (G). **(I, J)** Basal and maximum (max) $Ca^{2+}$ concentrations in peroxisomes based on (I). **(K)** Measurement with pericam-px according to the paradigm in (E), n = 27 cells from three experiments. **(L)** Simultaneous measurement of cytosolic (blue) and peroxisomal (green) $Ca^{2+}$. No delay of signal increase after histamine addition, but a delayed drop of the signal in peroxisomes. Left y-axis: of D3cpV-px (peroxisomal sensor). Right y-axis: $F_n/F_0$ ratio of R-GECO1 (cytosolic sensor), n = 35 cells from three experiments. **(L, M)** Decline of $F_n/F_0$ ratio per minute (min) in the linear part of the curves in (L) (from second 65–115, t test). Kinetic delay in decrease in peroxisomal signal is seen. **(A, B, C, G, H, I, K, M)** Data presented as mean ± SEM. **(I)** Data presented as Tukey's box plots. ***$P < 0.001$.

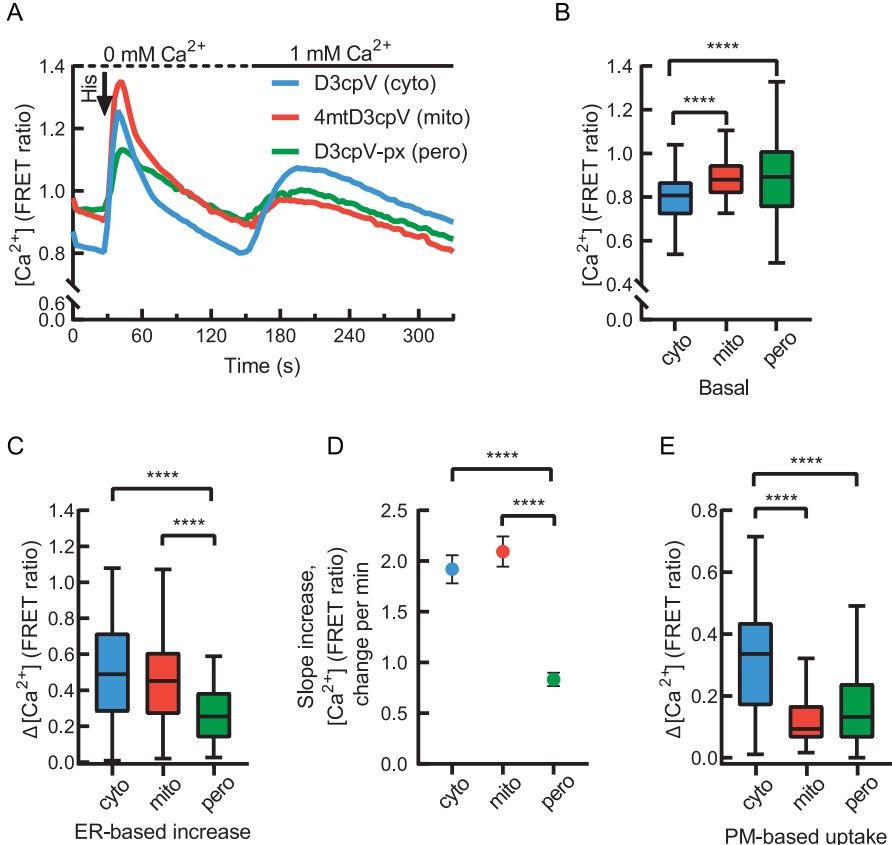

**Figure 3. Comparison of peroxisomal with cytosolic and mitochondrial Ca²⁺ handling in HeLa cells.**
**(A)** Comparison of cytosolic, mitochondrial, and peroxisomal Ca²⁺ response measured following the two-step measurement described in Fig 1E. Characteristic two peaks present in all three compartments. **(A, B)** Basal levels of Ca²⁺ in peroxisomes are similar to mitochondria. Analysis performed based on the data from (A). **(C)** Peroxisomal Ca²⁺ increase upon ER-store depletion is smaller than that of cytosol or mitochondria. Analysis performed based on the data from (A). **(A, D)** Increase of Förster resonance energy transfer ratio per minute (min) in the linear part of the curves in (A) (from second 27–42). Slower increase in peroxisomal signal is seen. Data presented as mean ± SEM. **(E)** Peroxisomal Ca²⁺ increase upon plasma membrane-based cellular uptake of Ca²⁺ is comparable to mitochondria. Analysis performed based on the data from (A). **(B, C, D, E)** One-way ANOVA followed by Tukey's post hoc test was used for the statistical analysis. ****$P < 0.0001$, Cyto: cytosolic, mito: mitochondrial, pero: peroxisomal. n = 83 (cyto), 116 (mito), 117 (pero) cells from six independent experiments. **(B, C, E)** Data presented as Tukey's box plots.

the current settings (Fig 3B). Furthermore, the increase in Ca²⁺ in peroxisomes upon intracellular store depletion with 100 μM histamine was significantly lower than the increase in the cytosol or mitochondria (Fig 3C). The Ca²⁺ increase rate in peroxisomes was more than twice lower than in cytosol or mitochondria (Fig 3D), speaking against the hypothesis that peroxisomal Ca²⁺ is rising drastically upon stimulation as suggested before (Lasorsa et al, 2008). The addition of extracellular Ca²⁺ resulted in another peak in all three compartments (Fig 3E), evidencing that peroxisomes, like mitochondria, depend on the PM-based uptake. Altogether, this suggests that peroxisomes tend to follow Ca²⁺ dynamics of the cytosol.

### Peroxisomal Ca²⁺ is not regulated by the mitochondrial calcium uniporter (MCU) complex

We examined the possible influence of mitochondrial Ca²⁺ uptake on peroxisomal Ca²⁺ by performing knockdown of the main component and pore forming part of the MCU complex (De Stefani et al, 2011) with siRNA (Fig 4). Knockdown efficiency assessed by qPCR was more than 90% (Fig 4A). A non-targeting siRNA (siCtrl) was used as a control. Ca²⁺ measurements with R-GECO-mito as a mitochondrial Ca²⁺ sensor showed reduced Ca²⁺ uptake in MCU knockdown after histamine addition compared with the control (Fig 4B). The significantly decreased Ca²⁺ uptake to mitochondria further confirms the reduction in MCU activity in the knockdown (Fig 4C). In same

cells co-expressing D3cpV-px, peroxisomes responded to histamine addition with signal increase in both control and knockdown (Fig 4D). We observed no difference in the Ca²⁺ uptake to peroxisomes immediately after histamine treatment (Fig 4E). However, after the peak peroxisomal Ca²⁺ remained constant in MCU knockdown, whereas the signal decreased gradually in the control. The observed increase could be attributed to an additional Ca²⁺ load. Less Ca²⁺ enters mitochondria in the MCU knockdown and the excess could enter peroxisomes. Together these results suggest that MCU is not responsible for Ca²⁺ transport across the peroxisomal membrane but peroxisomes may function as Ca²⁺ buffering system that takes up excess Ca²⁺. Moreover, these findings suggest that peroxisomal and mitochondrial Ca²⁺ homeostases are tightly interconnected.

### Peroxisomal Ca²⁺ in cardiomyocytes rises with cytosolic Ca²⁺ increase

We next tested in neonatal rat cardiomyocytes (NRCMs) the hypothesis that Ca²⁺ can access cardiac peroxisomes. NRCMs are primary cells with a well-developed T-tubule system and serve as a model for electrophysiological studies on CMs (Soeller & Cannell, 1999; Morad & Zhang, 2017).

We adapted the chemical stimulation protocol for the CMs by reducing it to a single stimulation because the main source of Ca²⁺ in these cells is the ER. We used thapsigargin (Tg) to chemically

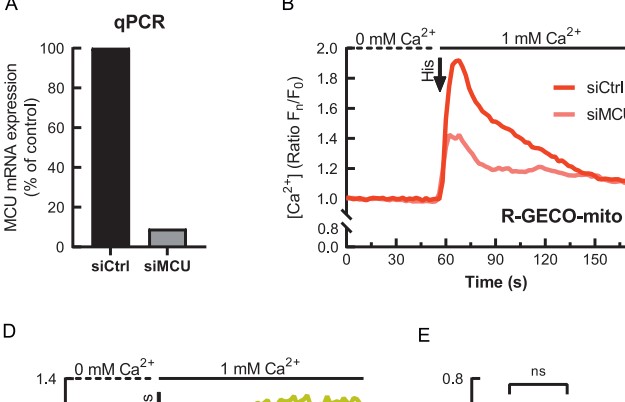

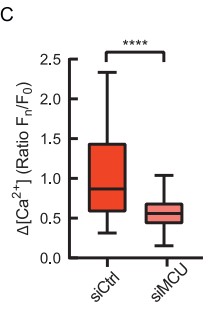

**Figure 4. Effect of mitochondrial calcium uniporter (MCU) knockdown on mitochondria and peroxisomal Ca²⁺ measured in the same cells.**
**(A)** mRNA expression of MCU-silenced (siMCU) HeLa cells in comparison to control (siCtrl), quantified by RT-qPCR. **(B)** Mitochondrial Ca²⁺ response in siCtrl and siMCU measured after the histamine addition with Ca²⁺-containing buffer. MCU knockdown results in decreased Ca²⁺ uptake to mitochondria. **(C)** Mitochondrial Ca²⁺ increase is smaller in siMCU. **(B)** Analysis performed based on the data from (B). **(D)** Peroxisomal Ca²⁺ response in siCtrl and siMCU measured following the histamine addition with Ca²⁺-containing buffer. **(E)** MCU knockdown does not affect maximum Ca²⁺ uptake to peroxisomes. **(D)** Quantification of data from (D). Cell numbers n = 73 (siCtrl), 61 (siMCU). **(C, E)** Data presented as Tukey's box plots. ****$P < 0.0001$.

stimulate the CMs (Fig 5A). To avoid measurement distortion by spontaneous contractile activity of CMs, cells were treated with 2,3-butanedione monoxime (BDM) (Gwathmey et al, 1991) before the experiment. As a proof of concept and for direct comparison, we performed the first round of measurements using the cytosol-localized Ca²⁺-sensor D3cpV (Fig 5B–D). Comparison between Tg treatment and buffer alone (Fig 5B) demonstrated, as expected, no differences in the basal ratios (Fig 5C), but an increase in cytosolic Ca²⁺ upon Tg addition (Fig 5D).

To measure peroxisomal Ca²⁺ changes, we transfected NRCMs with D3cpV-px and compared Tg treatment with the untreated control group (Fig 5E). No offset of basal ratios between the two groups was present before treatment (Fig 5F). After the addition of the SERCA inhibitor peroxisomal Ca²⁺ increased, evidencing peroxisomal Ca²⁺ uptake in NRCMs after store depletion (Fig 5G).

In the next set of experiments, we wanted to know if peroxisomes of human cardiac cells are able to take up Ca²⁺. To test this, human iPSCs created from fibroblasts of a healthy donor were differentiated into CMs using standardized protocols (Fig 5H). The possibility to generate hiPSCs from somatic cell sources and to direct their differentiation into almost any cell type make it possible to maintain and study human CMs in culture (Yoshida & Yamanaka, 2011). Cardiac differentiation was tested for homogeneity by using the cardiac specific marker cardiac troponin T (cTNT) and analysis by flow cytometry at day 90 of differentiation. Our differentiations yielded 90–95% cTNT-positive cells (Fig 5I). Staining of hiPSC-CMs with antibodies against α-actinin showed a regular sarcomeric striation pattern (Fig 5J).

As a proof of concept and for direct comparison, we measured cytosolic and peroxisomal Ca²⁺ and compared Tg treatment with the addition of Ca²⁺-free buffer without Tg to the control cells (Fig 5K–P). Starting with the same basal ratios as the control samples (Fig 5K and L), Tg-treated hiPSC-CMs showed a Ca²⁺ increase after the treatment (Fig 5M).

After confirming that Tg can effectively deplete Ca²⁺ stores in hiPSC-CMs, we measured peroxisomal Ca²⁺ in these cells (Fig 5N).

No ratio differences were present before Tg treatment (Fig 5O). Ca²⁺-store depletion resulted in an increase in peroxisomal Ca²⁺, confirming peroxisomal Ca²⁺ uptake in hiPSC-CMs (Fig 5P). Altogether, these results suggest that peroxisomes in both, rat and human cardiomyocytes, are able to take up Ca²⁺ upon intracellular Ca²⁺-store depletion and cytosolic Ca²⁺ increase.

## Peroxisomal Ca²⁺ oscillates in cardiomyocytes

In contrast to non-excitable cells, cell depolarization is the main stimulus for the initiation of Ca²⁺ signalling in CMs. The action potential depolarizes the cell membrane resulting in the activation of voltage-gated L-type Ca²⁺ channels (LTCC) in T-tubules (Chapman, 1979; Bootman et al, 2002). As a result, an initial small amount of Ca²⁺ enters the cell, activating RyRs on the sarcoplasmic reticulum membrane, resulting in Ca²⁺ release from the stores. This Ca²⁺-induced Ca²⁺ release enables cardiac muscle contraction. During relaxation, SERCA and NCX (sodium-calcium exchanger) pump Ca²⁺ back to the intracellular Ca²⁺ stores and out of the cells (Clapham, 2007).

We performed a series of stainings to visualize relative localisation of peroxisomes and D3cpV-px to the LTCC and RyR in hiPSC-CMs (Fig 6). Both, stainings with anti-Pex14 antibody and transfection with D3cpV-px revealed that peroxisomes are occasionally in contact with LTCC (Fig 6A). More often proximity of peroxisomes was detected to ER-resident RyR2 (Fig 6B). These findings are in accordance with the knowledge that peroxisomes make contact sites with the ER (Costello et al, 2017; Hua et al, 2017).

To enable more physiological interpretation of Ca²⁺ entry to peroxisomes in a beat-to-beat manner in NRCMs, we field-stimulated the cells with 1 Hz frequency (Fig 7A–D). Under field stimulation, we observed rhythmic changes of Ca²⁺ level in the cytosol (Fig 7A). To quantify the amplitude of changes and link to the stimulation, we performed fast Fourier transformation (FFT) of the data (Fig 7B). Signal amplitude

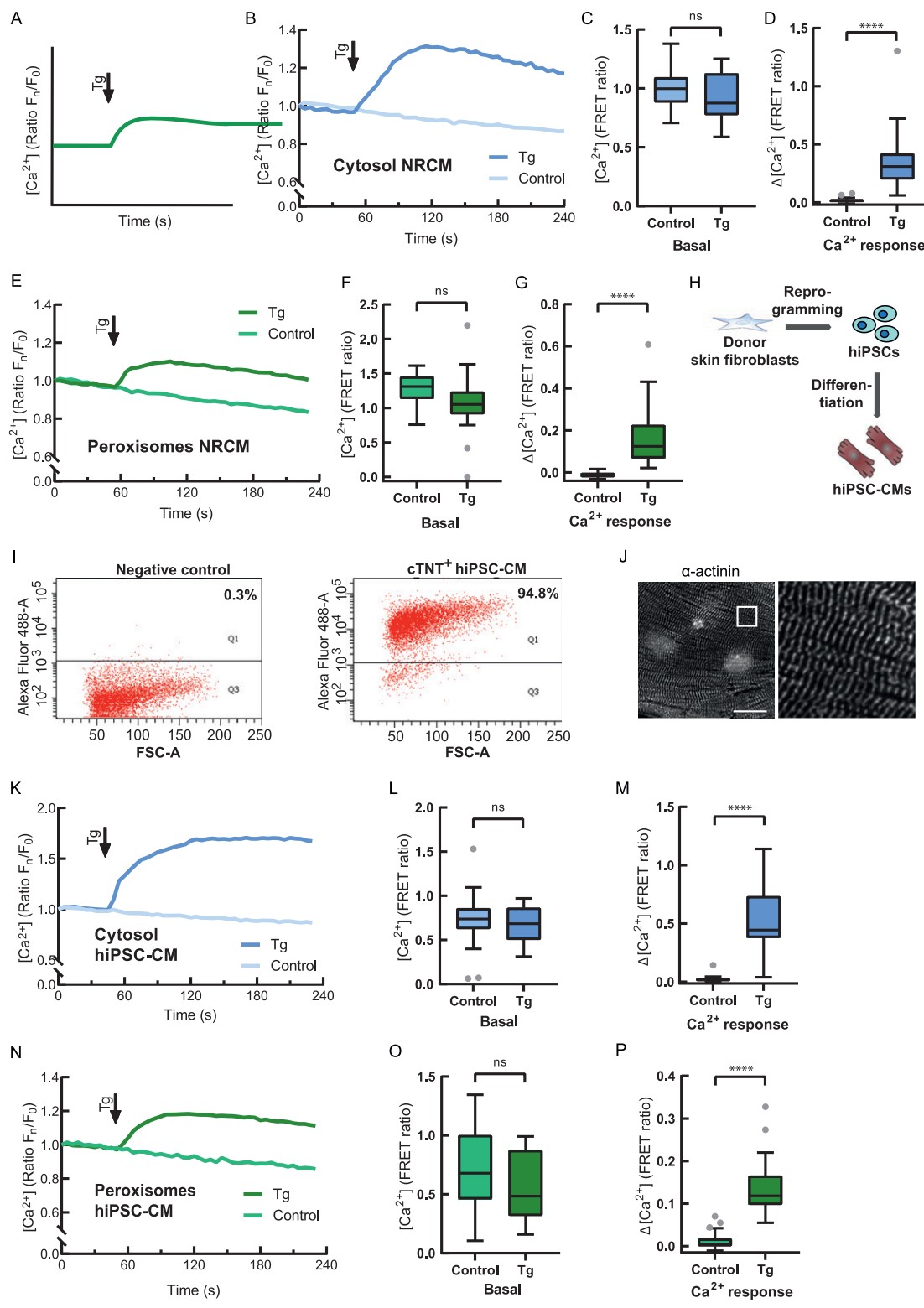

**Figure 5. Measurement of peroxisomal Ca$^{2+}$ in cardiomyocytes.**
**(A)** Experimental paradigm of Ca$^{2+}$ measurement in excitable cells. The peak after thapsigargin (Tg) addition represents Ca$^{2+}$ increase due to the sarcoplasmic/endoplasmic reticulum calcium ATPase inhibition and Ca$^{2+}$ retention in the cytosol. **(A, B)** Cytosolic Ca$^{2+}$ measurement in NRCMs following the experimental design in (A), n = 25 (Tg), 22 (control) from three experiments. Addition of Tg is compared with the addition of Tg-free buffer (control). **(B, C)** Basal levels are not different before the treatment in (B). **(B, D)** After Tg addition in (B) cytosolic Ca$^{2+}$ increases. **(A, E)** Peroxisomal Ca$^{2+}$ measurement in NRCMs following the experimental design in (A). Addition of Tg is compared with the addition of Tg-free buffer (control), n = 20 (Tg), 31 (control) from three experiments. **(E, F)** Basal levels of Ca$^{2+}$ are not different before the

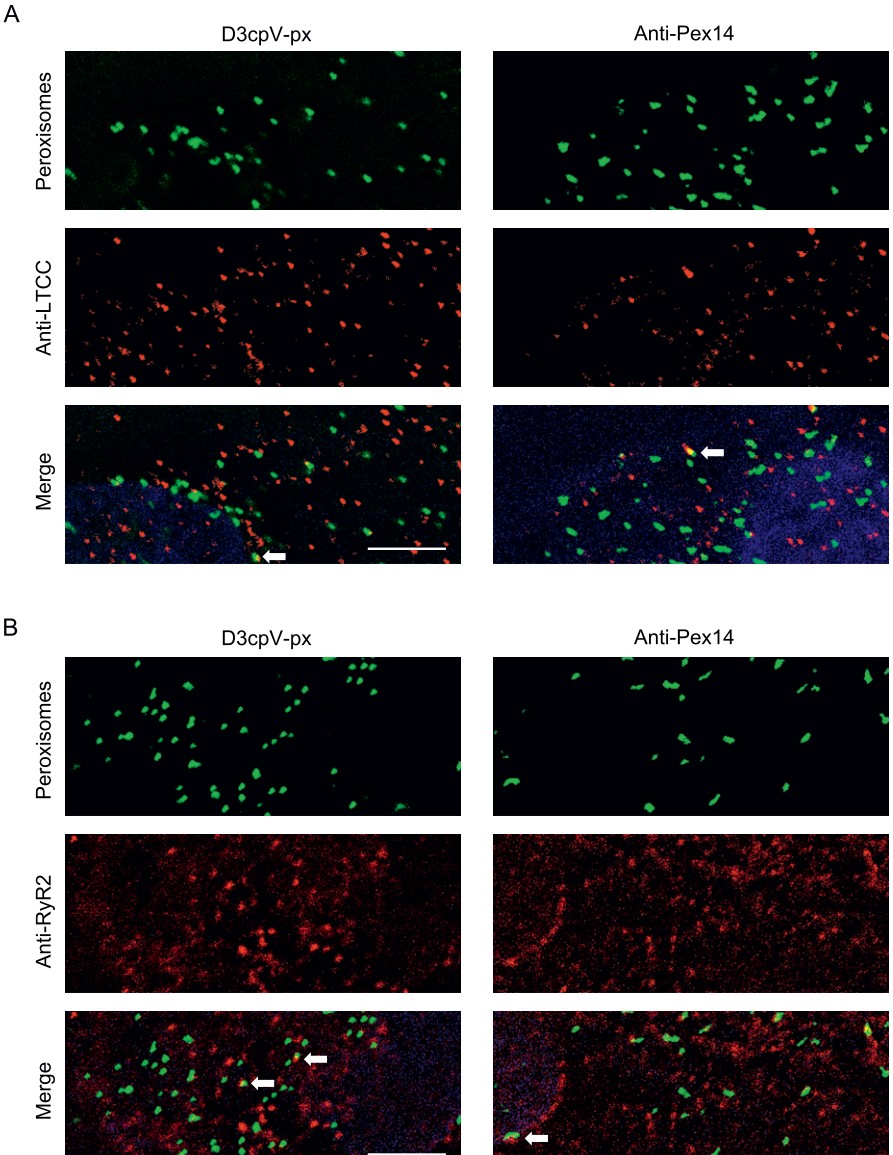

A

| Peroxisomes | D3cpV-px | Anti-Pex14 |
| Anti-LTCC | | |
| Merge | | |

B

| Peroxisomes | D3cpV-px | Anti-Pex14 |
| Anti-RyR2 | | |
| Merge | | |

**Figure 6.  Relative localisation of peroxisomes and Ca²⁺ channels in human induced pluripotent stem cell–CMs.**

Human-induced pluripotent stem cell–CMs were either transfected with D3cpV-px (left panels) or stained with anti-Pex14 (right panels) as a peroxisomal marker. **(A)** Representative images from staining of L-type Ca²⁺ channel (LTCC) show occasional proximity of peroxisomes and LTCC. **(B)** Representative images from staining of ryanodine receptor (RyR2) show occasional yet more often contact of peroxisomes with the RyR2 than with LTCC. DAPI is shown in blue. Scale bar 5 $\mu$m.

oscillations in the cytosol were rhythmic and corresponded to the stimulation frequency (Fig 7B).

To test peroxisomal response to electrical stimulation, NRCMs expressing D3cpV-px were paced at a frequency of 1 Hz (Fig 7C). Oscillations observed were smaller in amplitude and appeared less regular than the cytosolic responses. To identify the frequency domain of these oscillations we performed FFT (Fig 7D). The extracted pattern showed amplitude changes at 1 Hz, suggesting that peroxisomes take up Ca²⁺ in beat-to-beat manner. Together, our results suggest that peroxisomal Ca²⁺ in CMs is dependent on excitation-contraction process.

treatment in (E). **(E, G)** Peroxisomal Ca²⁺ increases after Tg addition in (E). **(H)** Human-induced pluripotent stem cell (HiPSC)-CMs generation. Donor skin fibroblasts were reprogrammed to hiPSCs, which were then differentiated to CMs. **(I)** hiPSC-CMs were stained for cardiac troponin T (cTnT) and analysed by flow cytometry. Negative control without primary antibody. 94.8% of iPSC-CMs are cTnT-positive (cTNT⁺). **(J)** Immunofluorescence staining visualized α-actinin protein expression and regular sarcomeric organisation. Scale bar: 20 $\mu$m. **(A, K)** Cytosolic Ca²⁺ measurement in hiPSC-CMs with D3cpV following the experimental paradigm for excitable cells in (A). Addition of Tg is compared to the addition of Tg-free buffer (control) to avoid artefacts and false results of the mechanical effect on the cells because of the addition itself. n = 24 (Tg), 27 (control) from three experiments. **(K, L)** No difference is found between two groups before the treatment in (K). **(K, M)** Tg addition in (K) results in cytosolic Ca²⁺ increase. **(A, N)** Peroxisomal Ca²⁺ measurement in hiPSC-CMs with D3cpV-px following the experimental design for excitable cells depicted in (A). Addition of Tg is compared with the addition of Tg-free buffer (control). n = 26 (Tg), 33 (control) from three experiments. **(N, O)** Basal levels of Ca²⁺ are not different before the treatment in (N). **(M, P)** Peroxisomal Ca²⁺ increases after Tg addition in (M). **(B, E, K, N)** Data presented as means from three independent experiments. **(C, D, F, G, L, M, O, P)** Unpaired $t$ test was used for the statistical analysis. ****$P$ < 0.0001, Tukey's box plots.

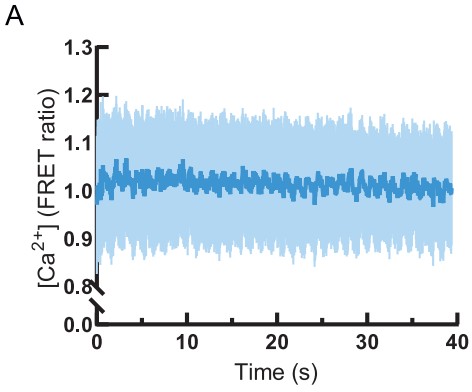

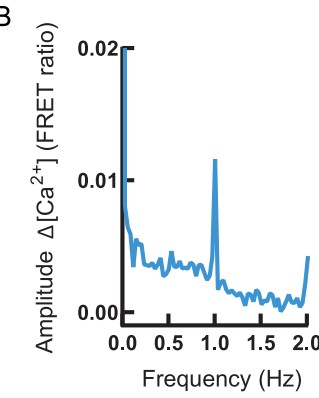

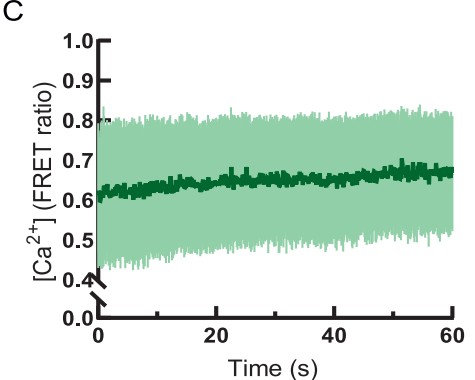

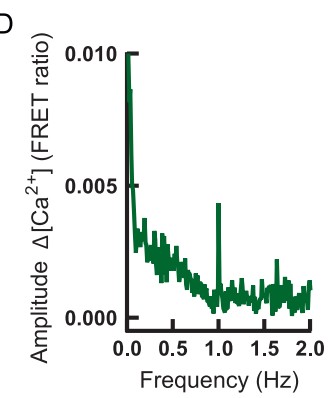

**Figure 7. Measurement of peroxisomal Ca²⁺ in paced cardiomyocytes.**
**(A)** D3cpV transfected NRCMs are stimulated with 1 Hz. Images are taken every 50 ms. Oscillations of Förster resonance energy transfer ratio are seen, n = 3.
**(A, B)** FFT from the data in (A). Signal increases are rhythmic and correspond to the pacing frequency.
**(C)** Förster resonance energy transfer ratio oscillates in D3cpV-px transfected NRCMs stimulated with 1 Hz. Images are taken every 100 ms, n = 3. **(C, D)** FFT from the data in (C). Signal increases are rhythmic and correspond to the pacing frequency.

## Discussion

Peroxisomes are metabolically highly active organelles in need of communication with other cellular compartments (Sargsyan & Thoms, 2020). ROS signalling and homeostasis are central to the participation of peroxisomes in signalling pathways (Lismont et al, 2019). In the present work, we focused on Ca²⁺ dynamics of peroxisomes as one of the major signalling molecules in the cell. We demonstrate that Ca²⁺ can enter peroxisomes of HeLa cells both when ER-stores are depleted and when cytosolic Ca²⁺ increases after Ca²⁺ entry across the PM.

Two articles published in 2008 brought forth conflicting data on peroxisomal Ca²⁺. According to Drago et al (2008), the basal level of Ca²⁺ in peroxisomes equals the cytosolic Ca²⁺ level, whereas Lasorsa et al (2008) find peroxisomal Ca²⁺ to be 20 times higher than in the cytosol. Whereas Lasorsa et al (2008) report rise of peroxisomal Ca²⁺ up to 100 μM using an aequorin-based sensor, Drago et al (2008) suggest slow increase when cytosolic Ca²⁺ rises. Each of the groups used a single yet different technique. These differences in the results can be partially attributed to the different measurement methods and the cell types used. Aequorin imaging requires long incubation times and cell population-based analysis that can be disadvantageous when measuring Ca²⁺ in intracellular organelles. In our experiments with HeLa cells, we found sixfold higher basal peroxisomal Ca²⁺ level than in the cytosol and increase up to 2.4 μM upon stimulation (Table 1). The range of the changes we report are based on the measurements with D3cpV-px and are supported by the measurement with pericam-px. Hence, we

conclude that D3cpV-px can be used for measuring peroxisomal Ca²⁺ concentration in a broad variety of cell types.

Electron microscopic experiments on rodent hearts performed in the 1970s show that peroxisomes are closely associated with T-tubules and with junctional sarcoplasmic reticulum (Hicks & Fahimi, 1977). We show that peroxisomes in CMs localise more often in ER vicinity than to T-tubular system. The sarcoplasmic reticulum is an indispensable site for the excitation-contraction coupling and Ca²⁺ handling in myocytes (Flucher et al, 1994). The localisation of peroxisomes to these sites raises the question if cardiac peroxisomes react to Ca²⁺ oscillations on a beat-to-beat basis, and/or if they can buffer calcium. HiPSC-CMs provide a wide spectrum of possibilities in cardiac research ranging from drug screening to cardiac regeneration (Yoshida & Yamanaka, 2011). In addition, these cells have been especially used to study patient-specific disease models including arrhythmic disorders and cardiomyopathies demonstrating a robust correlation to the predicted phenotype (Borchert et al, 2017; Prondzynski et al, 2019). We report here that Ca²⁺ is entering peroxisomes upon intracellular Ca²⁺-store depletion in rat and human CMs. Because intracellular store depletion is the main source of Ca²⁺ in CMs in the process of excitation-contraction coupling, it can be hypothesized that peroxisomes take up Ca²⁺ also in beat-to-beat manner in these cells.

Indeed, measurement of peroxisomal Ca²⁺ in CMs with FRET sensors in field stimulation confirms that peroxisomal Ca²⁺ increases in beat-to-beat manner. This suggests that peroxisomes may participate in excitation-contraction processes. The exact role of peroxisomes here is the matter of future research. Furthermore, the experimental

protocols with chemical stimuli developed here can be applied to study peroxisomal $Ca^{2+}$ in other cell types such as neurons.

We found that basal peroxisomal $Ca^{2+}$ levels are higher than cytosolic levels. There are two major ways of generating this $Ca^{2+}$ gradient on the two sides of the membrane. One option could be the energy-dependent uptake mechanism, such as SERCA for the ER (Clapham, 2007). We are, however, not aware of any data that can support this model. The second option may be locally high $Ca^{2+}$ concentration at the entry site that would allow more direct channeling of $Ca^{2+}$ (from the ER) into the peroxisomes resulting in relatively high peroxisomal $Ca^{2+}$. This second mechanism is known from the mitochondrial $Ca^{2+}$ handling, where ER–mitochondria contact sites with tethering proteins generate microdomain with locally high $Ca^{2+}$ concentration (Hirabayashi et al, 2017). As a result, $Ca^{2+}$ entry to mitochondria follows the $Ca^{2+}$ gradient but mitochondrial $Ca^{2+}$ is higher than the cytosolic $Ca^{2+}$. For the plausibility of the second option for peroxisomes speak the existence of ER–peroxisome contact sites (Costello et al, 2017; Hua et al, 2017). Therefore, we propose a hypothetical model of this mechanism (Fig 8), which, if true, will kick-start the search for the molecular identity of its components.

Although peroxisomal $Ca^{2+}$ levels are higher than cytosolic levels, peroxisomes are unlikely to store significant amounts of $Ca^{2+}$ under normal conditions, and they themselves take up $Ca^{2+}$ when intracellular stores are depleted. Under specific conditions, like apoptosis or oxidative stress, the situation may change, however. We observe gradual peroxisomal $Ca^{2+}$ increase in the case of MCU knockdown. This provides the first hint that under specific conditions peroxisomes may function as additional $Ca^{2+}$ buffer and take up excessive $Ca^{2+}$ that may harm the cells. We show that the rise of peroxisomal $Ca^{2+}$ after histamine stimulation is not delayed and largely follows the cytosolic $Ca^{2+}$. Although there could be a delay due to the binding and conformational changes of GECIs needed before the detection of the increase in the FRET signal, the range of this delay is less than milliseconds and cannot be seen in the experiments described here. We conclude that peroxisomes respond to cytosolic $Ca^{2+}$ because we only found concordant changes of $Ca^{2+}$ concentration in these two compartments.

The question of the cellular function and potential targets of peroxisomal $Ca^{2+}$ is still open. One of the roles of $Ca^{2+}$ could be the regulation of peroxisomal processes. On the other hand, metabolic processes themselves may regulate $Ca^{2+}$ uptake to organelles, as known from mitochondria (Nemani et al, 2020). A mutual regulation of metabolic pathways or ROS production localised to peroxisomes can be suggested based on the fact that $Ca^{2+}$ channel blockers nifedipine and diltiazem have suppressive effect on peroxisomal enzymes (Watanabe & Suga, 1988). Peroxisome proliferator-activated receptors system may be the connecting point between the metabolic processes, ROS, peroxisome abundance, and cellular $Ca^{2+}$ homeostasis (Colasante et al, 2015).

Some catalases from plant but not mammalian catalases can bind $Ca^{2+}$ (Yang & Poovaiah, 2002). Currently, there are no peroxisomal processes known in mammals that would directly depend on $Ca^{2+}$. Peroxisomes, however, could serve as an additional cytosolic buffer for $Ca^{2+}$ to take up an excess of cytosolic $Ca^{2+}$ and release it slowly. Based on the findings of this study that the $Ca^{2+}$ concentration in the peroxisome is higher than in the cytosol, it could be that peroxisomes may also serve as additional $Ca^{2+}$ source for the cytosol in extreme situations. The buffering function of peroxisomes may thus be important in the pathogenesis of arrhythmias.

## Materials and Methods

### DNA constructs

D3cpV-px (PST 1738) was generated from (pcDNA-)D3cpV (kind gift from A Palmer and R Tsien [Palmer et al, 2006] [#36323; Addgene]) by amplifying an insert with OST 1599 (GCGCATCGAT GGTGATGGCC AAGTAAACTA TGAAGAG) and OST 1600 (GCGCGAATTC TTAGAGCTTC GATTTCAGAC TTCCCTCGA) primers. The product was then reinserted into D3cpV using ClaI and EcoRI restriction sites. (pcDNA-)4mtD3cpV was a kind gift from A Palmer and R Tsien (Palmer et al, 2006) (#36324; Addgene). D1cpV-px (PST 2169) was generated from the (pcDNA-)D1cpV (Palmer et al, 2004) (#37479; Addgene) by amplifying an insert with oligonucleotide OST 2003 (GCGCGGATCC CATGGTGAGC AAGGGC) and OST 2002 (CGCGGAATTC TTAGAGCTTC GATTTCAGAC TTCCTATGAC AGGCTCGATG TTGTGGCGGA TCTTGAAGTT). The product was then reinserted into D1cpV using EcoRI and BamHI restriction sites. Pericam-px (PST 2170) was generated from ratiometric pericam (for mitochondria) (Nagai et al, 2001) by amplifying an insert with OST 2116 (GCGCAAGCTT ATGAAGAGGCGC TGGAAGAAAA) and OST 2117b (GCGCGAATTC CTAGAGCTTC GATTTCAGAC TTCCTATGAC AGGCTTTGCT GTCATCATTT GTA-CAAACT), which was then reinserted into ratiometric pericam using EcoRI and HindIII restriction sites. (CMV-)R-GECO1 and mito-R-GECO (#46021; Addgene) were kind gifts from R Campbell (Zhao et al, 2011). EYFP (Clontech) and Venus-PTS1 (PST1226) were used as the acceptor control in two-step measurements.

**Figure 8. Peroxisomal $Ca^{2+}$ entry and cellular $Ca^{2+}$ distribution.**
ER $Ca^{2+}$ release triggers $Ca^{2+}$ entry into the peroxisome. In this hypothetical model, ER–peroxisome proximity defines $Ca^{2+}$ microdomains with locally elevated $Ca^{2+}$ concentration shielded from the cytosol. As a result, $Ca^{2+}$ entry to peroxisomes follows the local gradient but peroxisomal $Ca^{2+}$ is eventually higher than in the cytosol. IP3Rc, $IP_3$ receptor calcium release channel of the ER.

For cloning of pVenus-PTS1 (PST1226), oligonucleotides OST801 (CACCCCTGTC ATAGGAAGTC TGAAATCGAA GCTCTAG) and OST802 (CTAGAGCTTC GATTTCAGAC TTCCTATGAC AGGGGTG) encoding the C-terminal decapeptide ACOX3 were annealed and cloned into the pENTR/D-TOPO cloning vector, yielding pENTR-ACOX3dp (PST1209). The resulting insert was transferred to pDEST-Venus using the Gateway cloning system.

## Cells, cell culture, and immunofluorescence

HeLa cells were cultured in low glucose DMEM medium (Biochrom) supplemented with 1% Pen/Strep (100 U/ml penicillin and 100 µg/ml streptomycin), 1% (wt/vol) glutamine, and 10% (vol/vol) FCS in 5% $CO_2$ at 37°C. For immunofluorescent analysis, cells were fixed with 4% paraformaldehyde for 30 min, and permeabilized using 1% Triton X-100 in PBS for 10 min. After blocking for 30 min with 10% BSA in PBS (blocking buffer) at 37°C, antigens were labelled with primary antibodies at room temperature for 1 h. Rabbit anti-PEX14 (ProteinTech) and mouse anti-PMP70 (Sigma-Aldrich) primary antibody dilution in blocking buffer was 1:500, and 1:200 for mouse anti-RyR (Invitrogen) and goat anti-LTCC (Santa Cruz). Labelling with the secondary antibodies conjugated to Cy3 (Life Technologies), Alexa Fluor 488 (Life Technologies), Alexa Fluor 633 (Invitrogen), or Alexa Fluor 647 (Jackson ImmunoResearch) was done for 1 h (1:500). Cover slips were mounted with ProLong Gold mounting medium with or without DAPI (Thermo Fisher Scientific). Images were taken with Axio Observer Z1 (equipped Zeiss Colibri 7 and with 63× oil Fluar) and deconvoluted. Colocalisation analysis was performed with Fiji (http://fiji.sc/) according to ImageJ User Guide. Mandor's colocalisation coefficient was measured after applying MaxEntropy thresholding on images.

NRCMs were isolated from newborn rats. Briefly, after the rats were euthanized, hearts were removed from the thoracic cavity, homogenized mechanically and digested in 1 mg/ml collagenase type II containing calcium- and magnesium-free PBS at 37°C with magnetic stirring. Supernatant was taken every 20 min and transferred to DMEM medium supplemented with Glutamax (Thermo Fisher Scientific), 10% FCS and 1% Pen/Strep. Cells were then centrifuged, the cell pellet resuspended in fresh medium and transferred to a Petri dish for 45 min (37°C and 5% $CO_2$). The fibroblasts adhered and NRCMs remained in the supernatant. NRCMs were then seeded on glass cover slips covered by Geltrex (Thermo Fisher Scientific).

Cells and cardiac differentiation of hiPSCs using standardized protocols, including cardiac mesoderm induction by subsequent activation and inhibition of the WNT pathway (Lian et al, 2013) and metabolic selection (Tohyama et al, 2013) were described earlier (Borchert et al, 2017). Cells were studied 90 d after initiation of differentiation. After differentiation, purity of hiPSC-CMs was determined by flow cytometry analysis (>90% cardiac TNT[+]) or by morphology (Borchert et al, 2017). HiPSC-CMs were maintained in RPMI 1640 supplemented with Glutamax, Hepes and B27 supplement.

## siRNA-mediated protein knockdown and qPCR

Transient knockdown was generated using siRNA from Microsynth (siMCU_1 sense: 5′-CAG GUG CCU UGC AAA GGU UGA–dTdT 3′; siMCU_2 sense: 5′-CUG GUC AUU AAU GAC UUA ACA dTdT-3′; siCtrl

sense: 5′ UUC UCC GAA CGU GUC ACG U - dTdT 3′). 3 million cells were transfected by nucleofection (Amaxa Nucleofector; Lonza GmbH) using the SE Cell Line Kit (#V4XC-1012) according to manufacturer's instructions with 4 µl of a 20 µM siRNA stock solution (for MCU using 2 µl siMCU_1 and 2 µl siMCU_2). All measurements were performed 72 h after transfection and the knockdown efficiency was confirmed using the same cells for qPCR.

## RT-qPCR

Total isolated RNA (800 ng) was reverse transcribed to cDNA using SuperscriptIV (#18090050; Thermo Fisher Scientific) and 1 µl was used for RT-qPCR using the GoTaq qPCR Master Mix (#A6002; Promega) and Realtime PCR System (Stratagene-Mx3000P; Agilent). TBP (TATA box–binding protein) was used as a housekeeping gene. Data were analysed using the $2^{-\Delta CT}$ method. Primers used: TBP_forw: 5′-CGGAGAGTTCTGGGATTGT-3′, TBP_rev: 5′-GGTTCGTGGCTCTCTTATC-3′, MCU_A_forw: 5′-CACACAGTTTGGCATTTTGG-3′, MCU_A_rev: 5′-TGTCTGTCTCTGGCTTCTGG-3′.

## Ca$^{2+}$ measurements

Cells (200,000 for HeLa and hiPSC-CMs and 500,000 for NRCMs) were seeded on glass cover slips and transfected with sensor plasmids using Effectene (QIAGEN) (HeLa) or Lipofectamine LTX Reagent (Thermo Fisher Scientific) (hiPSC-CMs and NRCMs) according to the manufacturers' instructions. Cells were imaged using a Zeiss Observer D1 (equipped with a EC-Plan Neofluar 40×/1.3 Oil Ph3 objective; Axiocam 702 mono and LED system Colibri; Zeiss) or Axio Observer Z1 (equipped with 40×/1.3 Oil Fluar objective, Zeiss Axiocam 702, Definite Focus.2 and Zeiss Colibri 7) at 37°C in a Ca$^{2+}$-free imaging buffer (145 mM NaCl, 4 mM KCl, 10 mM Hepes, 10 mM glucose, 2 mM $MgCl_2$, and 1 mM EGTA, pH 7.4 at 37°C) 24 h (HeLa and NRCMs) or 48 h (hiPSC-CMs) after transfection. Where indicated, NRCMs were field-stimulated at 1 Hz with MyoPacer ES (IonOptix). Data were analysed with AxioVision (Zeiss) and ZEN (Zeiss) software. Background and bleed-through (BT) were corrected in the FRET/donor ratio:

$$\frac{FRET}{donor} = \frac{(FRET - background) - [(CFP - background) \times BT] - [(YFP - background) \times BT]}{CFP - background}$$

Excitation 420 ± 20 and 505 ± 15 nm with emission filters 483 ± 16 and 542 ± 14 nm, or excitation 438 ± 12 and 508 ± 11 nm with emission filters 479 ± 20 and 544 ± 14 nm were used. For R-GECO measurements excitation was 550 ± 16 nm and emission 630 ± 46 nm. Where indicated, the concentration of Ca$^{2+}$ in the imaging buffer was increased to 1 mM by doubling the buffer volume to the cells (e.g., during treatment with chemicals) by the addition of Ca$^{2+}$-containing buffer (imaging buffer that contains 2 mM $CaCl_2$ [pH 7.4, 37°C] instead of EGTA).

The apparent dissociation constant $K_d$ value in the experimental setup was determined based on the titration protocol described for cytosolic GECI (Park & Palmer, 2015) with some modifications. Briefly, cells were washed with Ca$^{2+}$-, Mg$^{2+}$-, and EGTA-free buffer (pH 7.2) and incubated in Ca$^{2+}$- and Mg$^{2+}$-free buffer (pH 7.2) containing 3 mM EGTA and 5 µM ionomycin until the FRET signal reached

its minimum $R_{free}$ (for both D3cpV and D3cpV-px 5–6 min). D3cpV-px expressing cells were additionally incubated in 0.01% digitonin, 3 mM EGTA, and 5 µM ionomycin for 50 s then washed with $Ca^{2+}$-, $Mg^{2+}$-, and EGTA-free buffer (pH 7.2). Cells were immediately exposed to different $Ca^{2+}$ concentrations (once per experiment) and the $R_{final}$ was acquired. Buffers with different $Ca^{2+}$ concentrations were made as described previously (Park & Palmer, 2015). The results were fitted in a one-site model with Hill coefficient using GraphPad Prism 9 software. The procedures for the calculation of absolute $Ca^{2+}$ concentration were described earlier (Palmer & Tsien, 2006).

HiPSC-CMs and NRCMs were incubated in 10 mM 2,3-butanedione monoxime (BDM) before the measurements. FRET ratios (calculated as FRET donor ratio) were calculated by subtracting the background intensity and correcting for crosstalk. ER-store depletion in the cells was induced by 100 µM histamine (HeLa) in $Ca^{2+}$-free buffer or 1 µM Tg in $Ca^{2+}$-containing buffer. For permeabilisation, cells were treated with 0.01% digitonin in $Ca^{2+}$-free EGTA buffer for 50 s to 1 min and cytosol was washed out by rinsing twice with $Ca^{2+}$-free EGTA buffer. Cell response to ionomycin was measured by the addition of 5 µM ionomycin in 10 mM $Ca^{2+}$-containing buffer. Images for color LUT were made by applying Royal LUT on difference image of FRET and CFP in case of D3cpV-px and D1cpV-px, or difference image of 505 and 420 nm in case of pericam-px.

### pH control experiments

The preparation of buffers for the experiments is described elsewhere (Godinho & Schrader, 2017), with the exception that $Ca^{2+}$ in the buffers here was substituted with 1 mM EGTA. Before the experiments, cells were incubated in the $Ca^{2+}$-free imaging buffer, which was then removed. After washing the cells once with the according pH buffer, fresh buffer containing 10 µM nigericin was added to the cells. Images were taken with the same settings as for $Ca^{2+}$ measurements but every 15 s. After the stabilisation of the FRET ratio values the imaging was continued for another two-three minutes from which the mean value of five to seven time points was taken.

### Statistical analysis

Statistical significance was assessed using two-sided unpaired $t$ test when comparing two groups, or one-way ANOVA followed by Tukey's post hoc test when three groups were compared. Data were presented as Tukey's box plots: the box is limited by 25[th] and 75[th] percentiles. Data points larger than 75[th] percentile plus 1.5 IQR (interquartile range) or smaller than 25[th] percentile minus 1.5 IQR are presented as outliers. The extreme outliers were excluded from the graphical presentation of data. The whiskers cover all other data.

## Supplementary Information

## Acknowledgements

We thank Drs Robert Campbell, Takeharu Nagai, and Nicolas Demaurex for providing genetically encoded calcium indicators for optical imaging (GECO) and pericam plasmids. We thank Julia Hofhuis for earlier work on peroxisomal calcium and for cloning of D3cpV-px, Xin Zhang for support with microscopy, Roman Tsukanov for building the fluidics system for pH measurements, Johanna Heine for excellent technical assistance, and Matthias Plessner for comments on the manuscript. This project was supported by grants from the Deutsche Forschungsgemeinschaft TH 1538/3-1 to S Thoms, the Collaborate Research Council "Modulatory units in heart failure" SFB 1002/2 TP A10 to S Thoms and SFB1190 TP17 and SFB1027 TP C4 to I Bogeski, the MWK/VW foundation Project 131260/ZN2921 to S Thoms, the Horst and Eva-Luise Köhler Foundation to S Thoms, the Fritz Thyssen Foundation Az 10.19.2.026MN and IRTG1816 to K Streckfuss-Bömeke, and a PhD stipend by the DAAD program 57381412 ID 91572398 to Y Sargsyan. This work was supported by the DZHK (German Centre for Cardiovascular Research).

## Author Contributions

Y Sargsyan: formal analysis, investigation, visualization, and writing—original draft, review, and editing.
U Bickmeyer: investigation and writing—review and editing.
CS Gibhardt: investigation.
K Streckfuss-Bömeke: resources, supervision, and writing—original draft.
I Bogeski: resources, supervision, and writing—original draft.
S Thoms: conceptualization, resources, formal analysis, supervision, funding acquisition, visualization, project administration, and writing—original draft, review, and editing.

### Conflict of Interest Statement

The authors declare that they have no conflict of interest.

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
