## [Reviewer comments · Life Science Alliance]

Life Science Alliance

Peroxisomes contribute to intracellular calcium dynamics in cardiomyocytes and non-excitabile cells

Yelena Sargsyan, Uta Bickmeyer, Christine Gibhardt, Katrin Streckfuss-Bömeke, Ivan Bogeski, and Sven Thoms

DOI: <https://doi.org/10.26508/lsa.202000987>

Corresponding author(s): Sven Thoms, Medical School Bielefeld University

Review Timeline:

Submission Date:	2020-12-11
Editorial Decision:	2020-12-22
Revision Received:	2021-04-22
Editorial Decision:	2021-06-09
Revision Received:	2021-06-29
Editorial Decision:	2021-07-02
Revision Received:	2021-07-09
Accepted:	2021-07-12

Transaction Report:

December 22, 2020

Re: Life Science Alliance manuscript #LSA-2020-00987-T

Dr. Sven Thoms
University Medical Center
Department of Child and Adolescent Health
Robert-Koch-Str. 40
Robert-Koch-Strasse 40
Göttingen 37075
Germany

Dear Dr. Thoms,

Thank you for submitting your manuscript entitled "Peroxisomes contribute to intracellular calcium dynamics in HeLa cells and cardiomyocytes" to Life Science Alliance.

The manuscript was submitted and reviewed via Review Commons. The authors then chose to transfer their somewhat revised manuscript, along with the reviewers' comments and a proposed revised plan to Life Science Alliance (LSA). The reviewer comments and revision plan was assessed at LSA, and LSA editors deemed that the manuscript could be further considered at LSA provided the authors revise the manuscript, in accordance to what they have laid out in the pbp rebuttal / revision plan.

We, thus, encourage you to submit a revised manuscript to us that includes all the experiments you have laid out in their Revision plan, including the experiment with a pH-sensor in the peroxisome. Given that new data will be added to the revised manuscript, the revision will have to be looked at by a set of referees, most likely the same ones as Review Commons.

Thank you for this interesting contribution to Life Science Alliance. We are looking forward to

receiving your revised manuscript.

Sincerely,

Shachi Bhatt, Ph.D.
Executive Editor
Life Science Alliance
<https://www.lsjournal.org/>
Tweet @SciBhatt @LSAJournal

- A letter addressing the reviewers' comments point by point.
- An editable version of the final text (.DOC or .DOCX) is needed for copyediting (no PDFs).
- High-resolution figure, supplementary figure and video files uploaded as individual files: See our detailed guidelines for preparing your production-ready images, <https://www.life-science-alliance.org/authors>
- Summary blurb (enter in submission system): A short text summarizing in a single sentence the study (max. 200 characters including spaces). This text is used in conjunction with the titles of papers, hence should be informative and complementary to the title and running title. It should describe the context and significance of the findings for a general readership; it should be written in the present tense and refer to the work in the third person. Author names should not be mentioned.

B. MANUSCRIPT ORGANIZATION AND FORMATTING:

Peroxisomes contribute to intracellular calcium dynamics in cardiomyocytes

Dear Reviewers, dear Editors,

Thank you for considering our manuscript. We found the reviewers' comments very helpful. They have guided us in this revision. We revised all parts of the manuscript and included new experiments. Please find below our detailed point-by-point response. We are looking forward to hearing from you and to publishing with you.

With kind regards,

Yelena Sargsyan and Sven Thoms

Reviewer #1 (Evidence, reproducibility and clarity (Required)):

Reviewer #1 (Significance (Required)):

These are straight forward studies aimed to develop probes to assess peroxisomal Ca²⁺ in rest and in response to receptor stimulation. The probes were designed to measure intraperoxisomal Ca²⁺ and the Ca²⁺ the peroxisome experience when cytoplasmic Ca²⁺ is increased. The probes fill a need in understanding peroxisomal Ca²⁺ and Ca²⁺ signaling in general and should be very useful to investigators in the field.

The comments are aimed to help in improving the studies and taking them to the next stage.

The grammar needs improvement and the introduction needs sharpening. It is long and, in many places, not to the point. The results and discussion sections are also quite verbose.

*In response to the reviewer's comment, we edited the manuscript accordingly, and restructured and rewrote the introduction entirely. For example, we shortened the iPSC introduction or the discussion of GECIs. In response to the other reviewers' comments, we included a few additional topics, such as the basal Ca²⁺ differences between peroxisomes and cytosol in the discussion.

The sidedness of the probes need to be validated further, especially since the peroxisomal Ca²⁺ increase follows the cytoplasmic and the slower reduction rate may result from the environment experienced by the probe. Simple experiments:
how the probes respond to Ca²⁺ ionophore;

*We now tested ionomycin as an ionophore. The results are included in the manuscript (page 4, lines 120-126) and presented in Figure 2C. The different kinetics of Ca²⁺ increase and decline in cytosol and peroxisome in the presence of the ionophore suggest that the cytosol and peroxisomes have a different Ca²⁺ regulation.

does Ca²⁺ reduced rapidly when removed from the media of the digitonin permeabilized cells?;

*We now included data showing digitonin permeabilization and cytosol wash-out of the cells transfected with peroxisomal GECI (page 3, lines 103-109; Figure 1G and H). We do not see any

significant change in Ca^{2+} . This gives additional evidence for the specificity of the sensors for peroxisomes and efficient targeting. The signal is clearly from within the peroxisome.

how the cytoplasmic and peroxisomal thapsigargin responses compare using the protocols in 2A and 4A?

*We performed this experiment as suggested from the reviewer (page 4, lines 127-131, Figure 2D). We observed only differences in the maximal signals from the cytosol and peroxisomes.

Sidedness of PEX13-D3cpV was not examined.

*The reviewer is here referring to the topology of the membrane sensor. We agree with the reviewer that a thorough investigation of the topology of the sensor would be required. As there is now conflicting evidence on the membrane topology of PEX13, and as the sensor expressed very poorly (precluding biochemical assessment of topology), we decided to remove the data and discussion of the peroxisome membrane-localized Ca^{2+} sensor from the manuscript. This does not affect the interpretation of the other experiments.

Calculation of peroxisomal Ca^{2+} are based on K_d reported in the literature. The K_d s of D3cpV-px and PEX13-D3cpV should be determined when in the peroxisome in permeabilized cells for the numbers to have any meaning.

*We took the K_d as reported in the literature. We think re-cloning the sensors into bacterial expression vectors, purifying the proteins from *E. coli*, and measuring the K_d in vitro would not be worthwhile. Lastly the targeting signals are a few amino acids at the C-terminus after a flexible linker (like a myc tag); it is very unlikely that this changes the binding properties of the sensors.

How the localization of the probes look in the differentiated cardiomyocytes? How it compares to RyRs, VACC, etc..

* We compared the localization of the Ca^{2+} probes and peroxisomes with RyR and L-type Ca^{2+} channels (LTCC) (page 7, lines 137-141, Figure 6). We noticed proximity/contact of peroxisomes to RyR and LTCC, which was slightly more often for the RyR.

The major weakness of the study is that the probes are used only as a tool. To enhance the study and bring it beyond an excellent technical achievement, the authors should use them to study a significant Ca^{2+} -dependent peroxisomal function and show how the use of the tools eliminate the role of Ca^{2+} in such a function.

*We agree that it would be nice to identify a Ca^{2+} -dependent function within the peroxisome. This is, however, not the topic of the study. It is important for us to state that this is not a methods paper. At the heart of the paper are scientific questions: Does Ca^{2+} enter peroxisomes under physiological conditions? How much Ca^{2+} is in peroxisomes? May peroxisomal Ca^{2+} be important for heart cells? We addressed these questions using the novel tools that we developed.

Thank you for the critical and helpful comment on our study and for acknowledging the need of a detailed assessment of peroxisomal Ca^{2+} .

Reviewer #2 (Evidence, reproducibility and clarity (Required)):

The manuscript by Sargsyan et al describes an unappreciated role for peroxisomes in Calcium dynamics. Specifically, the authors propose that GPCR/VDCC/SOCE-mediated cytosolic Ca²⁺ elevation is rapidly sensed by peroxisomes and sequestered. The authors used/generated a peroxisome-targeted genetically encoded Ca²⁺ indicators which is elegant and powerful tool to monitor the luminal Ca²⁺ dynamics.

****Comments:****

Peroxisomes are single membrane bound organelles which are conserved across species spanning from yeast to humans. While housing only ~100 proteins, they are responsible for essential steps in lipid metabolism, amino acid metabolism and ROS homeostasis. Unlike other organelles, peroxisomes import fully folded and cofactor-bound proteins into their matrix. Though peroxisomes house specific metabolic functions, there is extensive crosstalk with other organelles, including mitochondria. It is essential to test and define whether silencing/knockdown of mitochondrial Ca²⁺ transport components like MCU will impact peroxisome Ca²⁺ uptake upon stimulation with histamine or electrical stimulation.

Since peroxisomes buffer significant amount of Ca²⁺, it is worth testing whether blockade of mitochondrial Ca²⁺ uptake would not alter peroxisome mediated Ca²⁺ influx. This analysis will provide Ca²⁺ uptake rate of mitochondria vs peroxisomes (Mallilankaraman K. et al CELL 2012 and Nemani N. et al Science Signaling 2020).

**These are very fruitful suggestions that we now took in account in various parts of the manuscript. Starting from the physiological stimulation of ER-Ca²⁺ release that we used, we focused on the ER-derived Ca²⁺ signal. We performed siRNA-mediated knockdown of MCU. MCU depletion did not directly affect peroxisomal Ca²⁺ uptake after stimulation with histamine (page 5, lines 182-195, Figure 4). However, we noticed continuous Ca²⁺ increase in peroxisomes after the peak. We addressed this also in the Discussion (page 8, lines 305-307). It must be mentioned that an MCU-equivalent likely does not exist in the peroxisome based on published data on peroxisomal membrane proteins in the literature.*

Peroxisomal synthesis of plasmalogens is Ca²⁺ and oxygen tension dependent, it is essential to show that altering Ca²⁺ controls plasmalogen synthesis.

**This is possibly a rewarding direction for future research. While we are not aware of a step of plasmalogen synthesis in peroxisomes that could be enzymatically linked to intraperoxisomal Ca²⁺ requirement, and experimental assessment of this questions would require an independent manipulation of peroxisomal Ca²⁺ in the context of plasmalogen biosynthesis.*

In the introduction authors have stated that "Elevated mitochondrial uptake increases 39 mitochondrial reactive oxygen species (ROS) production and is associated with heart failure and ischemic 40 brain injury (Starkov et al., 2004; Santulli et al., 2015)." These cited articles remotely link MCU and ROS elevation. It is important to point out that Tomar et al 2016 Cell Reports clearly demonstrated that genetic ablation of MCU suppresses mROS production that is mitochondrial Ca²⁺ dependent.

*Thank you for this suggestion that further links ROS and mitochondrial calcium/MCU. Previously, in the introduction, we mentioned these data to motivate the focus on cardiomyocytes when studying organellar Ca^{2+} . We now cite the suggested papers in the introduction and discussion (pages 1-2, lines 35-36; page 8, lines 313-315), and we discuss the interdependence of mitochondrial ROS and Ca^{2+} (page x, lines x)

Reviewer #2 (Significance (Required)):

The significance of the work is very high. The authors employ a variety of complementary techniques and experimental systems to demonstrate that peroxisomes indeed buffer a large quantity of Ca^{2+} upon stimulation.

Thank you for this very positive evaluation of our study and the constructive suggestions!

Reviewer #3 (Evidence, reproducibility and clarity (Required)):

This study outlines calcium probes for assessing the poorly understood role of peroxisomes in calcium signaling. The authors suggest that these organelles sequester calcium from either calcium influx across the plasma membrane or from release from the ER/SR. This is important since we need to know more about the roles of these organelles in calcium homeostasis and signaling. However, it needs to be robustly demonstrated that the probes are targeted to the right organelle without confounding contamination from other organelles which can be very significant even for a small degree of mis-targeting.

****Major****

1. The difference between the signals seen between the peroxisome and cytosolic D3 versions are not compelling, other than a dampened spike with the former (higher resting levels, smaller peak). See below for pH concerns.

*The Ca^{2+} signals in peroxisomes and cytosol are indeed similar. However, there are marked differences:

1) the basal (resting) Ca^{2+} levels are different

2) the dynamics in Ca^{2+} influx and efflux are different thus resulting in a more prominent peak within the cytosol. To this end, we obtained very similar results with two different peroxisomal calcium sensors.

In the meanwhile, we used ionomycin as an ionophore to maximally increase the cytosolic calcium concentration. These results are now presented in Figure 2C. The different kinetic in Ca^{2+} increase and decline in cytosol and peroxisomes can also be observed in the presence of the ionophore. These results provide additional evidence that the peroxisomal sensor is shielded from the cytosol.

2. How clean is the peroxisome distribution? Prove that D3 spillover from its being partially in (or on) other compartments (e.g. cyto, ER) is not contributing to the changes. Selective manipulation of Ca^{2+} in these other compartments should not affect the peroxisome signal.

a. For example, the small changes in the D3-px could be explained by peroxisome not changing at all but rather the other compartments (where larger responses are observed) signal(s) contaminating the response.

*The reviewer raises an important question.

- We have quantitatively assessed a dozen of peroxisome targeting signals, from very weak to maximally strong. The signal used in this study is the strongest we could identify among PTS1 proteins, so amount of residual non-imported protein is minimized.
- In Figure 1D-F, we use digitonin-permeabilized cells that have lost their cytosol (containing potentially mistargeted sensor). The results from these experiments show that the Ca^{2+} response is largely coming from within the peroxisome.
- In Figure 1G-H we now included data showing signal changes during and after digitonin permeabilization and cytosol wash out of the cells transfected with peroxisomal GECl. We do not see any significant change in Ca^{2+} . This suggests that there is no signal loss with the cytosol and supports the idea of highly efficient targeting.
- We now tested the targeting by quantitating the localization (as suggested in minor comment 1 by the same reviewer). D3cpV-px localization to peroxisomes is comparable with peroxisomal membrane protein PMP70 localization with the organelle (page 3, lines 109-111, Figure 2A).
- The selective manipulations suggested by the reviewer, however, would affect peroxisomal calcium. In the manuscript we show that peroxisomal Ca^{2+} is dependent on cytosolic Ca^{2+} . A manipulation on the ER would affect cellular Ca^{2+} homeostasis since ER is the main Ca^{2+} store. The MCU knockdown that we performed shows that there could not be sensor mistargeting to mitochondria (page 5, lines 182-195, Figure 4).

b. e.g. if in the ER lumen, the signal should be eliminated with SERCA inhibitors (thapsigargin, CPA). They used Thapsigargin in cardiac myocytes, why not in HeLa during characterization)?

* Thank you very much for this suggestion. We now show the data of HeLa cell stimulation by thapsigargin (Tg) (page 4, lines 127-131, Figure 2D). We see signal increase in both cytosol and peroxisomes, suggesting that there is no mistargeting to the ER.

3. Any Ca^{2+} reporter will pH-sensitive to an extent, even D3 (Ca^{2+} binding, inherent fluorescent proteins).

a. It is essential to prove that the signal changes are not due changes perox pH. Target pH-sensitive proteins to the perox lumen by the same strategy and show that the same Ca^{2+} interventions do not cause pH changes.

* We agree, that it is important to rule out a misleading influence from pH sensitivity. The pH-sensitive component of the Ca^{2+} sensors is the YFP-based acceptor. We, therefore, performed the two-step histamine based experiment on the cells expressing only the acceptor (page 4, lines 140-144, Figure 2F). We do not see significant signal changes over the experiment, suggesting that the pH changes are not in the range to affect the YFP.

To exclude the effects of pH difference on the basal levels, we exposed the cells transfected with either D3cpV or with D3cpV-px to different pH buffers containing the proton ionophore nigericin, and EGTA but devoid of Ca^{2+} (page 4, lines 127-131, Figure 2B). We did not detect significant differences in the range of physiological cytosolic pH (7.2) and peroxisomal pH (usually reported between 7.0 and 8.0, depending on the cell type and source, (Godinho LF, and Schrader M. 2017. Determination of Peroxisomal pH in Living Mammalian Cells Using pHRed). Only the exposure of the sensors to pH 4.0 resulted in a steep decrease of FRET ratio.

We also considered the options of direct peroxisomal pH measurement with the experts in the field. Some pH sensors lose the pH-sensing properties when targeted to peroxisomes. The others are not suitable for the kinetic experiments over time (which is needed in the current case). The only (to our knowledge) published functional peroxisomal pH-sensor for kinetic measurements in undamaged cells (Godinho LF, and Schrader M. 2017. Determination of Peroxisomal pH in Living Mammalian Cells Using pHRed) could not be delivered due to a severe fire accident in the originating lab.

b. The authors claim different resting levels of $[Ca^{2+}]$ in cytosol/mitochondria/peroxisome. The resting FRET level also depends on the resting pH of the compartments which may also be different. Certainly, mitochondria are more alkaline than the cytosol. Again, to interpret these are real Ca^{2+} differences requires the pH to be accounted for.

*See above (Comment 1 and 3a by the same reviewer).

4. I am puzzled by the model, in particular in view of Fig 3. The genetically-encoded calcium indicator (GECI) is allegedly in on the cytosolic face of the peroxisome and measuring peri-peroxisomal Ca^{2+} .

a. The changes with this reporter look pretty similar to the luminal reporter (save that the resting ratio may be lower). I don't understand how the lumen $[Ca^{2+}] >$ cytosolic $[Ca^{2+}]$ without a higher local $[Ca^{2+}]$ (unless there is an energy-driven uptake mechanism, but then how does this fit in with ER-driven Ca^{2+} release?).

*Our data indeed show that the luminal signal responds to the cytosolic Ca^{2+} signal. The reviewer suggests that an energy-driven process is required to establish a concentration difference between luminal and the peroxisomal site. When looking at mitochondria, however, we could envision a (still hypothetical) model whereby a locally high Ca^{2+} concentration at the entry side allows the channeling of Ca^{2+} (from the ER) into the peroxisome. This model builds on the high Ca^{2+} concentration in the ER and neither requires an active import process nor an equilibration of cytosolic and peroxisomal Ca^{2+} concentrations. We elaborate on this model in the discussion (page 8, lines 290-301) and provide a schematic of the hypothetical Ca^{2+} domain at the peroxisome in Figure 8.

5. The claim that resting peroxisome $[Ca^{2+}]$ is higher than cytosol is questionable. Is this a calibration artifact (e.g. compartment pH-differences or the reporter behaves differently in the lumen)? Such a gradient could not be sustained without energy-dependent Ca^{2+} uptake. The authors make no discussion of this.

*As stated with the previous comment, if the ER-derived Ca^{2+} signal that we are measuring in our work, relied on ER-peroxisome proximity, domains at the cytosolic side of the peroxisome membrane would be expected to show increased local Ca^{2+} concentration (like at mitochondria). In addition, we have to consider that not only influx but the efflux also plays an important role in determining compartmental calcium concentrations. A gating mechanism of Ca^{2+} efflux would explain the different resting (steady state) concentrations. Additionally, it could explain the differences in decline rates that we find in cytosol and peroxisomes (Figure 2K and L). We address these points in the discussion (page 8, lines 290-301).

Minor

1. Quantitate localization. Pearson's coefficients for GECIs and Peroxisomes.

*We quantified the colocalization of GECI with peroxisomal marker catalase (page 3, lines 109-111, Figure 2A). See also Major comment 2a from the same reviewer.

2. Different upstroke rates of D3 with His vs Cao. Quantify.

*Thank you for this comment. We did this based on the data from Figure 3A (page 5, lines 176-179, Figure 3D). For that we calculated the increase of FRET/donor ratio per minute in the linear part of the curves from D3 Cyto, Mito, and Pero, similar to the way it was done in Figure 2L.

3. Page 5. Line 161. 'Different sites', do the authors mean different sides? Similarly, the Legend of Fig 3.

* Thank you for pointing this out. This part is not included in the new version of the manuscript.

Reviewer #3 (Significance (Required)):

Good peroxisome calcium probes is important to the general calcium signaling field. This is fundamental science of interest to all cell biologists.

There has been little published on peroxisome calcium, although for example, the Pozzan lab published a paper in JBC in 2008 on a GFP-based lumenally targeted peroxisome probe. There is contradictory data in the field and reliable new approaches are needed.

Thank you for critically questioning the technical details of this work and for highlighting the importance of this study.

June 9, 2021

Re: Life Science Alliance manuscript #LSA-2020-00987-TR

Prof. Sven Thoms
Medical School Bielefeld University
Department of Biochemistry and Molecular Medicine
Morgenbreede 1
Bielefeld 33615
Germany

Dear Dr. Thoms,

Thank you for submitting your revised manuscript entitled "Peroxisomes contribute to intracellular calcium dynamics in cardiomyocytes" to Life Science Alliance. The manuscript has been seen by the original reviewers whose comments are appended below. While the reviewers continue to be overall positive about the work in terms of its suitability for Life Science Alliance, one important issue remains regarding calculating the K_d in permeabilized cells as suggested by Reviewer 1.

Our general policy is that papers are considered through only one revision cycle; however, given that the suggested changes are relatively minor, we are open to one additional short round of revision.

Please submit the final revision within one month, along with a letter that includes a point by point response to the remaining reviewer comments.

-- Summary blurb (enter in submission system): A short text summarizing in a single sentence the study (max. 200 characters including spaces). This text is used in conjunction with the titles of papers, hence should be informative and complementary to the title and running title. It should

describe the context and significance of the findings for a general readership; it should be written in the present tense and refer to the work in the third person. Author names should not be mentioned.

B. MANUSCRIPT ORGANIZATION AND FORMATTING:

Sincerely,

Reviewer #1 (Comments to the Authors (Required)):

The authors addressed many of my comments but failed to address one issue. Determining the K_d for the probes while in the peroxisomes is essential for comparing basal and stimulated Ca^{2+} between compartments. My comment and the authors response are below. I did not suggest purifying the probe etc... The comment clearly indicates the need to permeabilize the cell to gain access to the peroxisomes and use various Ca^{2+} concentrations in the perfusate in the presence of Ca^{2+} ionophore to obtain the K_d when the probe is within the peroxisomes. Once this is determined and peroxisomal Ca^{2+} is calculated using these K_d s, the manuscript will be ready for publication.

Calculation of peroxisomal Ca^{2+} are based on K_d reported in the literature. The K_d s of D3cpV-px and PEX13-D3cpV should be determined when in the peroxisome in permeabilized cells for the numbers to have any meaning.

*We took the K_d as reported in the literature. We think re-cloning the sensors into bacterial expression vectors, purifying the proteins from *E. coli*, and measuring the K_d in vitro would not be worthwhile. Lastly the targeting signals are a few amino acids at the C-terminus after a flexible linker (like a myc tag); it is very unlikely that this changes the binding properties of the sensors.

Reviewer #2 (Comments to the Authors (Required)):

The authors have addressed the reviewer's comments adequately and the revised manuscript is suitable for publication.

Reviewer #3 (Comments to the Authors (Required)):

I previously supplied a detail report with concerns. The authors have carefully addressed these and

provided additional experimental data. This is particularly important with regard to possible pH effects. However, the authors have directly tested this and I am satisfied with their modifications and responses.

This paper is basically about the generation and characterization of a probe. It is more limited with regard to new biology.

Reviewer #1

The authors addressed many of my comments but failed to address one issue. Determining the K_d for the probes while in the peroxisomes is essential for comparing basal and stimulated Ca^{2+} between compartments. My comment and the authors response are below. I did not suggest purifying the probe etc... The comment clearly indicates the need to permeabilize the cell to gain access to the peroxisomes and use various Ca^{2+} concentrations in the perfusate in the presence of Ca^{2+} ionophore to obtain the K_d when the probe is within the peroxisomes. Once this is determined and peroxisomal Ca^{2+} is calculated using these K_d s, the manuscript will be ready for publication.

Calculation of peroxisomal Ca^{2+} are based on K_d reported in the literature. The K_d s of D3cpV-px and PEX13-D3cpV should be determined when in the peroxisome in permeabilized cells for the numbers to have any meaning.

*We took the K_d as reported in the literature. We think re-cloning the sensors into bacterial expression vectors, purifying the proteins from *E. coli*, and measuring the K_d in vitro would not be worthwhile. Lastly the targeting signals are a few amino acids at the C-terminus after a flexible linker (like a myc tag); it is very unlikely that this changes the binding properties of the sensors.

Answer: We thank the reviewer for this suggestion. We now performed the K_d calibration suggested by the reviewer (lines 146-150). For that we used the instructions for K_d calculation by the authors of the original construct D3cpV and performed the fitting as a one-site Hill coefficient model as suggested by the originators of D3cpV. Absolute calcium concentrations and calcium uptake were recalculated based on these figures.

Reviewer #2

The authors have addressed the reviewer's comments adequately and the revised manuscript is suitable for publication.

Answer: Thank you for the positive evaluation of our work.

Reviewer #3

I previously supplied a detail report with concerns. The authors have carefully addressed these and provided additional experimental data. This is particularly important with regard to possible pH effects. However, the authors have directly tested this and I am satisfied with their modifications and responses.

This paper is basically about the generation and characterization of a probe. It is more limited with regard to new biology.

Answer: Thank you for the careful evaluation of our contribution.

July 2, 2021

RE: Life Science Alliance Manuscript #LSA-2020-00987-TRR

Prof. Sven Thoms
Medical School Bielefeld University
Department of Biochemistry and Molecular Medicine
Morgenbreede1
Bielefeld 33615
Germany

Dear Dr. Thoms,

Thank you for submitting your revised manuscript entitled "Peroxisomes contribute to intracellular calcium dynamics in cardiomyocytes and non-excitable cells". We would be happy to publish your paper in Life Science Alliance pending final revisions necessary to meet our formatting guidelines.

- please consult our manuscript preparation guidelines <https://www.life-science-alliance.org/manuscript-prep> and make sure your manuscript sections are in the correct order
- please add your main figure and table legends to the main manuscript text after the references section
- please add a conflict of interest statement to your main manuscript text
- please include a separate Data Availability section if applicable

FIGURE CHECKS:

- missing scale bars for Figure 6B

A. FINAL FILES:

-- High-resolution figure, supplementary figure and video files uploaded as individual files: See our

detailed guidelines for preparing your production-ready images, <https://www.life-science-alliance.org/authors>

B. MANUSCRIPT ORGANIZATION AND FORMATTING:

Sincerely,

Reviewer #1 (Comments to the Authors (Required)):

My remaining concern has been addressed and the m/s should be accepted for publication.

July 12, 2021

RE: Life Science Alliance Manuscript #LSA-2020-00987-TRRR

Prof. Sven Thoms
Medical School Bielefeld University
Department of Biochemistry and Molecular Medicine
Morgenbreede1
Bielefeld 33615
Germany

Dear Dr. Thoms,

Thank you for submitting your Research Article entitled "Peroxisomes contribute to intracellular calcium dynamics in cardiomyocytes and non-excitable cells". It is a pleasure to let you know that your manuscript is now accepted for publication in Life Science Alliance. Congratulations on this interesting work.

DISTRIBUTION OF MATERIALS:

Again, congratulations on a very nice paper. I hope you found the review process to be constructive and are pleased with how the manuscript was handled editorially. We look forward to future exciting submissions from your lab.

Sincerely,
